# Curriculum Abductive Learning

**Wen-Chao Hu**[1,2]  **Qi-Jie Li**[1,2]  **Lin-Han Jia**[1]  **Cunjing Ge**[1,2]
**Yu-Feng Li**[1,2]  **Yuan Jiang**[1,2]  **Zhi-Hua Zhou**[1,2]

[1]National Key Laboratory for Novel Software Technology, Nanjing University, China
[2]School of Artificial Intelligence, Nanjing University, China
`{huwc,liqj,jialh,liyf,jiangy,zhouzh}@lamda.nju.edu.cn,gecunjing@nju.edu.cn`

## Abstract

Abductive Learning (ABL) integrates machine learning with logical reasoning in a loop: a learning model predicts symbolic concept labels from raw inputs, which are revised through abduction using domain knowledge and then fed back for retraining. Due to the nondeterminism of abduction, the training process often suffers from instability, especially when the knowledge base is large and complex, resulting in a prohibitively large abduction space. While prior works focus on improving candidate selection within this space, they typically treat the knowledge base as a static black box. In this work, we propose Curriculum Abductive Learning (C-ABL), a method that explicitly leverages the internal structure of the knowledge base to address the ABL training challenges. C-ABL partitions the knowledge base into a sequence of sub-bases, progressively introduced during training. This reduces the abduction space throughout training and enables the model to incorporate logic in a stepwise, smooth way. Experiments across multiple tasks show that C-ABL outperforms previous ABL implementations, significantly improves training stability, convergence speed, and final accuracy, especially under complex knowledge setting.

## 1 Introduction

Recent advances in data-driven machine learning methods have achieved remarkable success across a wide range of tasks [Zhou, 2021]. However, when it comes to leverage structured, formal knowledge—such as logic rules or domain-specific constraints—these methods often fall short, therefore mitigating their interpretability and reliability. This has motivated the emergence of data- and knowledge-driven artificial intelligence, which aims to tightly integrate learning from data with logical reasoning [Hitzler, 2022, Yang et al., 2025].

Abductive Learning (ABL) [Zhou, 2019, Zhou and Huang, 2022] is a representative framework that integrates machine learning with logical reasoning in a balanced loop. In ABL, a machine learning model first predicts symbolic concept labels from input data, which may initially violate domain knowledge when the model is under-trained; abduction then generates a set of concept label candidates compatible with domain knowledge, and from which the best candidate is selected via consistency optimization. The selected candidate is subsequently used to supervise the model's next update. This loop iterates, enabling the system to gradually align predictions with logical constraints.

While elegant in principle, this loop setup often introduces training inefficiency and instability [Yang et al., 2024, He and Li, 2025]. Due to the nondeterminism of abduction [Magnani et al., 2009], often multiple candidates compatible with the domain knowledge exist. When the knowledge base is complex, the resulting abduction space can become prohibitively large, causing the model to oscillate among many plausible yet incorrect concept labels. This undermines the reliability of the supervision signal and hinders the training process.

39th Conference on Neural Information Processing Systems (NeurIPS 2025).

Prior work [Huang et al., 2021, Tao et al., 2024, He et al., 2024] has attempted to improve ABL stability by refining consistency optimization, i.e., the selection process within the abduction space. While these methods offer incremental gains, they remain fundamentally limited when the space itself is too large. A key limitation is that they treat the knowledge base as a static black box, ignoring its internal structure [Hu et al., 2025]. We argue that a more principled solution requires **leveraging the structure of knowledge base and actively manage its complexity throughout training**.

Many knowledge bases exhibit inherent staged or hierarchical structural properties [Glanois et al., 2022]. Take the legal domain as an example [Huang et al., 2020], some laws and regulations are relatively simple and foundational, while others involve more complex conditions or exceptions. Such rules can be organized into phases of increasing complexity. Motivated by this commonly observed structure, we propose a logic-level **Curriculum Abductive Learning (C-ABL)** method. Instead of exposing the model to the full knowledge base from the beginning, C-ABL incrementally introduces logic rules over several phases during training. In this way, abduction space is substantially reduced throughout training, enabling faster and more stable optimization. As training progresses, increasingly complex logic is introduced, allowing for a gradual alignment with full-domain constraints.

To realize this idea, we first propose a principled algorithm that leverages the structural properties of knowledge base and partitions it into a sequence of sub-bases with strong local dependencies, increasing complexity, and self-contained reasoning. These are then integrated in ABL via a curriculum training process. Our theoretical analysis shows that this curriculum design achieves efficient and stable training within phases by leveraging prior knowledge learned in earlier phases, and ensures smooth transitions across phases without catastrophic forgetting. Experiments on digit addition, chess attack, and real-world legal judgment tasks demonstrate that C-ABL significantly improves training stability, convergence speed, and final accuracy over strong ABL and neuro-symbolic baselines.

In summary, our contributions are as follows:

- We identify the uncontrolled abduction space as the core bottleneck in ABL training and highlight the limitations of prior implementations.

- We propose the C-ABL method that explicitly structures and stages the logical knowledge base, transforming abduction from a static process into a dynamic and adaptive one.

- We theoretically and empirically demonstrate that our curriculum design leads to a more efficient and stable training across various domains.

To our knowledge, this is the first work to explicitly leverage the internal structure of the knowledge base in ABL—an essential yet previously overlooked component—shifting from black-box invocation to structure-aware, logic-enhanced training. This perspective opens new directions for improving ABL and extends its potential for broader and more reliable applications.

## 2 Problem Setting and Preliminary

### 2.1 Knowledge Base Concepts

We begin with a brief introduction to the basic concept of a **knowledge base**. This paper focuses on domain knowledge expressed in **logic programming** [Huth and Ryan, 2004, Ben-Ari, 2012], a formalism rooted in first-order logic that offers expressiveness and human interpretability, while remaining computationally tractable for reasoning. Specifically, the knowledge base $\mathcal{KB}$ is represented as a set of logic programming rules. Each **rule** is written in the Horn clause form [Horn, 1951]:

$$A \leftarrow B_1 \wedge \cdots \wedge B_n,$$

where "$\leftarrow$" denotes logical implication, meaning that $A$ (left-hand side, the **head** of the rule) holds when the conjunction of $B_i$ (right-hand side, the **body** of the rule) holds. Each $A$ and $B_i$ is a logical statement expressed with logical **predicates**.

Given such a knowledge base, both **deduction** and **abduction** [Josephson and Josephson, 1996, Magnani et al., 2009] can be performed. Deduction allows deriving $A$ (head) from the premises $\bigwedge_i^n B_i$ (bodies), while abduction seeks possible explanations $\bigwedge_i^n B_i$ (bodies) that would account for the observation of $A$ (head).

## 2.2 Problem Setting

The task of this paper is as follows: Given an input data sequence $\boldsymbol{x} = (x^{(1)}, \ldots, x^{(m)})$ of length $m$, where each element $x^{(i)}$ is sampled from an input space $\mathcal{X}$ and corresponds to a **concept label** $z^{(i)}$ drawn from a symbolic concept label set $\mathcal{Z} = \{z_1, \ldots, z_N\}$, the entire sequence $\boldsymbol{x}$ is associated with a **target label** $y$ that reflects a target reasoning outcome. In addition, we are provided with a logical knowledge base $\mathcal{KB}$, expressed in first-order logic, that defines how concept labels can entail specific target labels. Specifically, each concept label $z \in \mathcal{Z}$ appears as a predicate in the bodies of rules in $\mathcal{KB}$, and target labels $y$ appear in the heads.

The goal is to learn a model $f$ that maps each $x^{(i)}$ to a concept label $z^{(i)}$, such that the sequence $\boldsymbol{z} = (z^{(1)}, \ldots, z^{(m)})$, together with $\mathcal{KB}$, can be used to deduce the target label $y$, in other words, $\boldsymbol{z}$ satisfies logical condition $\boldsymbol{z} \wedge \mathcal{KB} \models y$.

**Example 2.1** (Decimal $d$-digit Addition). *Consider a task of predicting the sum of two decimal $d$-digit numbers, represented by a sequence of images [Manhaeve et al., 2018]. The input $\boldsymbol{x} = (x^{(1)}, \ldots, x^{(2d)})$ consists of $2d$ images from a visual input space $\mathcal{X}$, such as MNIST [LeCun et al., 1998] or CIFAR10 [Krizhevsky et al., 2009], (e.g., $\boldsymbol{x} = (\boxed{1}, \boxed{3}, \boxed{7}, \boxed{3})$)). The concept labels are drawn from the symbol set $\mathcal{Z} = \{\texttt{zero}, \ldots, \texttt{nine}\}$, and the target label $y$ is their sum. The knowledge base $\mathcal{KB}$ encodes the rules of multi-digit addition. The goal is to map each $x^{(i)}$ to its corresponding concept label $z^{(i)}$, such that the resulted sequence $\boldsymbol{z}$ (e.g., $[\texttt{one}, \texttt{three}, \texttt{seven}, \texttt{three}]$) and $\mathcal{KB}$ allows deduction of the correct sum $y$ (e.g., 86).*

## 2.3 Brief Introduction of Abductive Learning

Abductive Learning (ABL) [Zhou, 2019, Zhou and Huang, 2022] consists of two components: a machine learning model $f : \mathcal{X} \to \mathcal{Z}$ for concept perception, and a logical reasoning module that utilizes the first-order rules in $\mathcal{KB}$. During the training process, when the input $\boldsymbol{x}$ is received, ABL uses $f$ to map it to perceptive concept labels $\hat{\boldsymbol{z}}$, then the logical reasoning module checks whether $\hat{\boldsymbol{z}} \wedge \mathcal{KB} \models y$ holds, i.e., $\hat{\boldsymbol{z}} \wedge \mathcal{KB}$ can correctly deduce $y$. If the condition is not satisfied, ABL utilize the reasoning module to get revised concept labels $\bar{\boldsymbol{z}}$, and then, $\bar{\boldsymbol{z}}$ is used as the ground-truth concept label for $\boldsymbol{x}$ to train $f$. This process is repeated iteratively until the training of $f$ converge.

Specifically, the process by which the reasoning module obtains the revised concept labels $\bar{\boldsymbol{z}}$ involves the following two sequential steps [Dai et al., 2019, Huang et al., 2024]: (1) **Abduction**: Return the **abduction space**, i.e., the set of all concept label candidates that are compatible with $\mathcal{KB}$, denoted as $\mathbb{S} = \{\boldsymbol{z} \mid \boldsymbol{z} \wedge \mathcal{KB} \models y\}$; and (2) **Selection**: Performing consistency optimization to select the candidate $\bar{\boldsymbol{z}} \in \mathbb{S}$ that is most consistent with the model's current prediction $p = f(\boldsymbol{x})$:

$$\bar{\boldsymbol{z}} = \underset{\boldsymbol{z} \in \mathbb{S}}{\arg\min}\, 1 - \prod_{i=1}^{m} p^{(i)}(z^{(i)}), \tag{1}$$

where $p^{(i)}(z^{(i)})$ denotes the probability assigned by the model to label $z^{(i)}$.

**Challenges.** Due to the nondeterminism of abduction, for a given $y$, the size of abduction space $|\mathbb{S}|$ is often large, i.e., there may be multiple $\boldsymbol{z}$ satisfying $\boldsymbol{z} \wedge \mathcal{KB} \models y$. However, for each input $\boldsymbol{x}$, the correct concept labels should be unique [He et al., 2024]. Therefore, selecting an incorrect $\boldsymbol{z}$ from $\mathbb{S}$ introduces erroneous supervision to the perception model $f$, which could then be reinforced through subsequent iterations in the learning loop. This creates a vicious cycle that slows convergence, destabilizes training, and ultimately degrades final model accuracy.

**Example 2.2.** *In Example 2.1, given $y = 86$ and $\mathcal{KB}$, the resulting set $\mathbb{S}$ would be of size $L = 87$ ($\mathbb{S} = \{[\texttt{zero}, \texttt{zero}, \texttt{eight}, \texttt{six}], [\texttt{zero}, \texttt{one}, \texttt{eight}, \texttt{five}], \ldots$ ). However, the concept labels that are consistent with $\boldsymbol{x} = (\boxed{1}, \boxed{3}, \boxed{7}, \boxed{3})$ should be unique: $\bar{\boldsymbol{z}} = [\texttt{one}, \texttt{three}, \texttt{seven}, \texttt{three}]$. In this case, all other elements in $\mathbb{S}$ may introduce false labels that mislead the training of $f$.*

Several solutions have been proposed to address this training challenge, including methods by Huang et al. [2021], Tao et al. [2024], He et al. [2024], which primarily focus on consistency optimization, i.e., how to **select** the best candidate $\boldsymbol{z}$ within $\mathbb{S}$. However, such methods remain limited when $\mathbb{S}$ is large—no selection strategy can fully compensate for the challenges brought by an overwhelming hypothesis space [Yang et al., 2024, Jia et al., 2025]. Addressing the root of training instability requires going beyond improving candidate selection and actively managing the size and structure of the abduction space itself.

# 3 Our Method

This section proposes **Curriculum Abductive Learning** (**C-ABL**), a method that addresses the limitations in previous ABL methods. C-ABL actively managing the complexity of the knowledge base during training: Instead of exposing the model to the full knowledge base all at once, it divides ABL training into multiple curriculum **phases**, with each phase $p$ guided by a sub-base $\mathcal{KB}_p$, progressively expanding towards $\mathcal{KB}$.

## 3.1 Curriculum Design over Knowledge Base

To enable curriculum-style progression, we first introduce a principled algorithm that partitions the full knowledge base $\mathcal{KB}$ into a sequence of structured sub-bases $\mathcal{KB}_1, \ldots, \mathcal{KB}_P$ used in each curriculum phase. Our partitioning strategy follows three intuitive principles that together ensure each sub-base builds upon the last in a stable and constructive manner:

**P1 Dependency Cohesion** (*What should be grouped together?*): Rules depending on each other, such as those frequently co-occur or form tight reasoning chains, appear in the same phase.

**P2 Stepwise Complexity** (*What should be introduced first?*): Simpler rules are placed in earlier phases, while more complex rules built on prior predicates are introduced later.

**P3 Self-contained Reasoning** (*Can it reason independently?*): Each phase includes all rules necessary for reasoning over the involved concepts, without relying on unseen rules.

To implement these principles, we introduce the following structure:

**Definition 3.1** (Dependency Graph). Given a knowledge base $\mathcal{KB}$ consisting of first-order logic rules, its dependency graph $G = (V, E)$ is a directed graph where each node $r \in V$ corresponds to a rule, and there is an edge $(r_i, r_j) \in E$ if the head predicate of $r_i$ appears in the body of $r_j$.

This graph captures the structural reasoning flow: if $r_j$ relies on conclusions from $r_i$, there is an edge from $r_i$ to $r_j$. Based on this graph, we design a partitioning algorithm shown in Algorithm 1.

**Partitioning Algorithm.** The algorithm begins by scanning each concept label $z \in \mathcal{Z}$ (as introduced in Section 2.2, each $z$ is a predicate that appears in the body of some rule) and forms an initial cluster $C_z$ containing all rules that directly reference $z$ (Line 3). This cluster is then recursively expanded by traversing the dependency graph, incorporating additional rules that are required to derive the final target label $y$, as well as rules that define intermediate predicates used in this process (Line 4). In cases where the dependency graph contains cycles (e.g., formed by recursive rules), the above process will cluster all rules involved in the cycle together.

---

**Algorithm 1:** $\mathcal{KB}$ partitioning

**Input:** Knowledge base $\mathcal{KB}$ (a set of first-order logic rules), minimum sub-base size $\tau$ (optional)
**Output:** A sequence of sub-bases $\mathcal{KB}_1, \ldots, \mathcal{KB}_P$

1  Construct the dependency graph $G = (V, E)$ of $\mathcal{KB}$      $\triangleright$ from Definition 3.1;
2  **for** *each concept label $z \in \mathcal{Z}$* **do**
3      Initialize cluster $C_z$ with all rules referencing $z$;
4      Recursively expand $C_z$ via edges in $G$;
5  **for** *each pair $(C_a, C_b)$* **do**
6      **if** $C_a = C_b$ **then**
7         Merge clusters;
8      **else if** $\exists (r_i, r_j) \in E$ with $r_i \in C_a$, $r_j \in C_b$ **then**
9         Set $C_a \prec C_b$;
10 Topologically sort clusters under $\prec$;
11 Set phase counter $p \leftarrow 1$;
12 **for** *each sorted cluster $C_i$* **do**
13     If $|\mathcal{Z}_i| < \tau$, merge $C_i$ with the next cluster    $\triangleright$ $\mathcal{Z}_i$: the set of concept labels $C_i$ involves;
14     Assign sub-base: $\mathcal{KB}_p \leftarrow \bigcup_{j=1}^{i} C_j$;
15     $p \leftarrow p + 1$;

---

Once the initial clusters are formed, we refine them in two ways: (1) Merge any duplicate clusters with identical rule sets (Line 7); (2) Define a precedence ordering $\prec$, and perform topological sort (Line 9-10). Specifically, this ordering is based on the interdependencies of clusters: If there is an edge from a rule in $C_a$ to a rule in $C_b$, i.e., reasoning in $C_b$ relies on conclusions from $C_a$, we assign $C_a \prec C_b$ (when bidirectional dependencies exist, we let the cluster with fewer rules appear earlier).

Sub-bases are then formed incrementally by aggregating clusters up to a given index (Line 14). This process satisfies **P1** by grouping together rules in the same reasoning chain, keeping each phase logically localized around certain concept predicates. It also satisfies **P2**, as the precedence ordering introduces foundational rules and simpler concepts earlier, and allows subsequent phases to build upon prior ones, forming a progression from easy to complex reasoning throughout training.

To avoid overly fine-grained partitions, small clusters can be optionally merged with their immediate successors (Line 13). This step is governed by a optional threshold $\tau$, which specifies the minimum number of concept label predicates per phase; if not set, no merging is performed.

Illustrative examples of the above process are shown in Appendix B.1. We also provide the computational cost analysis in Appendix B.2, showing that compared to the ABL training pipeline, the partitioning is a one-time, offline preprocessing step with negligible computational overhead.

**Guarantee of Self-Contained Reasoning.** To ensure **P3**, we define the concept label domain of each phase $p$, denoted $\mathcal{Z}_p$, as the set of concept labels $\mathcal{KB}_p$ involves. We require that $\mathcal{KB}_p$ is logically sound and complete for reasoning over $\mathcal{Z}_p$, without relying on future rules. This is formalized below.

**Theorem 3.2.** *For any phase $p$, let $\varphi$ be any logical statement over concepts in $\mathcal{Z}_p$. Then:*

1. ***Soundness:*** *$\mathcal{KB}_p \models \varphi \implies \mathcal{KB} \models \varphi$; that is, $\mathcal{KB}_p$ does not derive any conclusions that are invalid under the full knowledge base.*

2. ***Completeness:*** *$\mathcal{KB}_p \models \varphi \impliedby \mathcal{KB} \models \varphi$; that is, $\mathcal{KB}_p$ is sufficient to derive all valid conclusions over its concept label domain.*

We then have the following corollary, indicating that the final sub-base $\mathcal{KB}_P$ is logically equivalent to the full knowledge base $\mathcal{KB}$ in terms of reasoning over all concept labels.

**Corollary 3.3.** *For the final phase $P$, for any formula $\varphi$ over $\mathcal{Z}$, we have: $\mathcal{KB}_P \models \varphi \iff \mathcal{KB} \models \varphi$.*

All proofs are provided in Appendix C.

## 3.2 Curriculum-Guided Training

Building on the sub-bases from Algorithm 1, we now present C-ABL, which conducts ABL training across $P$ curriculum phases. The pseudocode is provided in Algorithm 2 in Appendix D.

In phase $p$, training is guided by sub-base $\mathcal{KB}_p$. To align the model's prediction space with the reasoning scope of $\mathcal{KB}_p$, we dynamically schedule training data whose concept labels fall within the domain $\mathcal{Z}_p$ [Bengio et al., 2009, Marconato et al., 2023a]. Then, each phase follows the standard ABL procedure, as stated in Section 2.3.

Training in phase $p$ continues until the prediction accuracy for every label $z \in \mathcal{Z}_p$ exceeds the uniform guessing baseline $1/|\mathcal{Z}|$, and then C-ABL proceeds to the next phase $p + 1$. This is a mild criterion requiring only a weak signal for each concept to facilitate progression. Nonetheless, as shown in Section 4, reaching this level is sufficient for one phase to benefit learning in the next, enabling the model to build upon previously acquired knowledge in a curriculum-style progression. Note that concept labels are evaluated on a held-out validation set and are not used for training or backpropagation, and the only supervision signal during training comes from the target label $y$.

This process repeats until reaching the final phase $P$, at which point the full knowledge base $\mathcal{KB}$ is employed and training continues until a termination condition is met, such as reaching a maximum number of training iterations.

# 4 Theoretical Analysis: Efficient and Smooth Optimization

In this section, we provide a formal analysis of the Curriculum Abductive Learning (C-ABL) method. We demonstrate how C-ABL improves the training process of ABL by addressing two core aspects: (1) Improving the efficiency and stability of training within each phase, and (2) Ensuring smooth transitions across phases. All proofs are provided in Appendix C.

## 4.1 Efficient Training within Phase

We begin by analyzing each phase individually. In particular, we analyze how our method reduce the size of the abduction space by benefiting from curriculum training paradigm, and therefore contributes to more efficient reasoning and stable optimization.

The key bottleneck in ABL training arises from the size of the abduction space $\mathbb{S} = \{z \mid z \wedge \mathcal{KB} \models y\}$. This space $\mathbb{S}$ includes all possible concept labels compatible with the knowledge base, constituting the set of candidates ABL search and select from.

When using the full knowledge base, the size of $\mathbb{S}$ can grow exponentially:

**Lemma 4.1.** *If the knowledge base $\mathcal{KB}$ involves $N$ concept labels and the input has $m$ positions, then the size of the abduction space with full knowledge base is bounded by $|\mathbb{S}| \leq N^m$.*

To mitigate this, C-ABL partitions the knowledge base to reduce complexity. At phase $p$, the subset $\mathcal{Z}_p \subseteq \mathcal{Z}$ is active, and the abduction space is: $\mathbb{S}_p = \{z \in \mathcal{Z}_p^m \mid z \wedge \mathcal{KB}_p \models y\}$. The theorem below shows that, under the curriculum strategy proposed in Section 3.2, the size of $\mathbb{S}_p$ is largely reduced.

**Theorem 4.2.** *Assuming that all previously introduced concepts $z \in \mathcal{Z}_{p-1}$ are predicted with accuracy exceeding random chance (i.e., $Acc(z) > \frac{1}{|\mathcal{Z}|}$), and pseudo-labels are selected via Eq. (1). Then the size of the abduction space in phase $p$ is bounded by $|\mathbb{S}_p| \leq |\mathcal{Z}_p \setminus \mathcal{Z}_{p-1}|^m$.*

*Remark* 4.3. The above theorem relies on a natural assumption that the semantics of concepts remain constant when the knowledge base evolves (e.g., the image **3** consistently corresponds to the concept `three` regardless of training phase) [Marconato et al., 2023b, 2024]. This theorem suggests that the abduction space in each phase is effectively confined to newly introduced concepts $\mathcal{Z}_p \setminus \mathcal{Z}_{p-1}$, benefiting from the accumulated knowledge of earlier phases: The previously learned concepts provide a foundation with lower risk of noisy supervision [Van Krieken et al., 2024, Maene et al., 2024], enabling the model to focus on resolving the new concepts introduced in the current phase.

Combining Lemma 4.1 and Theorem 4.2, the relative upper bound of the phase-level abduction space satisfies

$$\frac{|\mathbb{S}_p|}{|\mathbb{S}|} \leq \left( \frac{|\mathcal{Z}_p \setminus \mathcal{Z}_{p-1}|}{|\mathcal{Z}|} \right)^m,$$

which shrinks exponentially in $m$. The following example illustrates this reduction:

**Example 4.4.** *In the $d$-digit addition task, as shown in Figure 1, the size of $\mathbb{S}$ increases rapidly with $d$ when applying the full knowledge base. In contrast, with phase partitioning (using a phase threshold $\tau = 2$ in Algorithm 1), the abduction space remains consistently small across all values of $d$.*

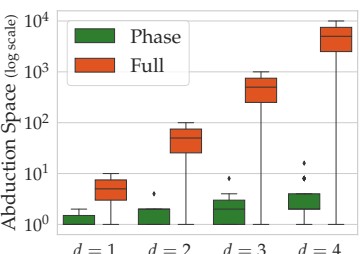

Figure 1: Comparision of abduction space w/ and w/o phase partitioning across different digit length.

This reduction yields two advantages:

1. **Improved computational efficiency:** A smaller abduction space significantly lowers the reasoning cost per training iteration, especially when the knowledge base is large, where the number of logically compatible candidates explodes exponentially, making abduction prohibitively expensive [Yu et al., 2016, Hu et al., 2025];

2. **Faster and stabilized convergence:** A large abduction space introduces many plausible yet incorrect candidates, especially in early training when the model's predictions are still noisy. These incorrect candidates can mislead training and cause the model to oscillate or reinforce suboptimal hypotheses. Limiting the abduction space will promote faster, more stable convergence, which is captured in the following theorem:

**Theorem 4.5** (Iteration Complexity of ABL). *Let $f^{(t)}$ be the model at iteration $t$ in ABL. Assuming that: (1) At each iteration, the model receives a pseudo-label $\bar{z}^{(t)} \in \mathbb{S}$ that satisfies $\bar{z}^{(t)} \wedge \mathcal{KB} \models y$; (2) The model updates are driven by minimizing a loss function $\ell(f(x), \bar{z}^{(t)})$ that is convex and $\rho$-Lipschitz; (3) The learning rate $\eta > 0$ is fixed. Then, the number of iterations $T$ required to reach an expected consistency error less than $\varepsilon$ satisfies: $T = \mathcal{O}\left(\frac{|\mathbb{S}|^2 \cdot \rho^2}{\varepsilon^2}\right)$.*

This theorem formally shows that the number of required iterations grows quadratically with the size of the abduction space. Therefore, reducing $\mathbb{S}$ in each phase not only improves the speed of logical reasoning but also accelerates overall convergence, particularly in large knowledge base scenarios.

### 4.2  Smooth Phase Transition

Besides improvements within each phase, we now examine transitions across phases. Two properties are key to maintaining stability: **logical consistency**, ensuring conclusions made in earlier phases remain valid, and **topological continuity**, ensuring the optimization landscape evolves with no abrupt changes. Together, these properties help prevent catastrophic forgetting [Bengio et al., 2009], a common challenge in curriculum learning where new updates may destabilize previously acquired knowledge, by preserving previously acquired knowledge both semantically and parametrically as the curriculum progresses.

To ensure logical consistency, we require that conclusions derived in one phase remain valid in all subsequent phases. The following theorem shows sub-bases from Algorithm 1 satisfy this property:

**Theorem 4.6.** *For any formula $\varphi$ over $\mathcal{Z}_p$, we have $\mathcal{KB}_p \models \varphi$ if and only if $\mathcal{KB}_{p+1} \models \varphi$.*

To analyze topological continuity, we adopt the notion of Stone spaces [Johnstone, 1982], which offer a topological view of logical model sets (a brief introduction is provided in Appendix C.7). Assuming each concept label $z_i$ is a unary predicate with a Boolean assignment, the set of formulas over $\mathcal{Z}_p$ then forms a Boolean algebra $\mathcal{B}_p$. Its corresponding Stone space $S(\mathcal{B}_p)$—the set of ultrafilters over $\mathcal{B}_p$—captures the space of logically consistent models. For these model spaces, we have:

**Theorem 4.7.** *For each $p$, $S(\mathcal{B}_p) \subseteq S(\mathcal{B}_{p+1})$.*

As a result, the model spaces form a nested sequence of compact topological spaces, ensuring that admissible models evolve smoothly, without causing abrupt shifts in the optimization landscape.

## 5  Experiments

In this section, we evaluate our method across three tasks. On digit addition and its variants, the most widely used benchmarks in the neuro-symbolic field, we thoroughly validate the effectiveness of our method. On a chess attack task, we test robustness when only Boolean target label is provided, leading to a significantly larger abduction space. Finally, we apply our method to a real-world legal judgment task to demonstrate its practical applicability in complex domains.

### 5.1  $d$-digit Addition

This task, as outlined in Example 2.1, the input consists of image sequences representing two $d$-digit numbers, and the goal is to predict their sum. We aim to assign a concept label to each image, so that the final result is obtained through reasoning. The standard decimal setting includes 10 concept labels, and we use images from CIFAR [Krizhevsky et al., 2009] to represent digits. To extend the challenge, we introduce a hexadecimal variant with 16 concept labels, constructed using the first 16 classes of CIFAR100 [Krizhevsky et al., 2009]. More details and additional experimental results are provided in Appendix F.1.

We compare our method against two ABL baselines: (1) the original **ABL** [Huang et al., 2024], and (2) the improved variant **A³BL** [He et al., 2024]. We also include several representative neuro-symbolic methods: (1) **NeurASP** [Yang et al., 2020], (2) **LTN** [Serafini and Garcez, 2016], (3) **DeepProbLog** [Manhaeve et al., 2018, 2021a], and (4) **DeepStochLog** [Winters et al., 2022]. Introduction of compared methods are provided in Appendix E. All methods use ResNet18 [He et al., 2016] as the perception module, and are trained for a total of 5,000 iterations.

Table 1: Comparison of prediction accuracy (top) and total training time in minutes (bottom) on digit addition tasks. "N/A" indicates runtime exceeding 12 hours. C-ABL consistently achieves higher accuracy and faster training, with its benefits more pronounced as reasoning complexity increases.

| Method | Decimal | | | | Hexadecimal | | |
|---|---|---|---|---|---|---|---|
| | $d=1$ | $d=2$ | $d=3$ | $d=4$ | $d=1$ | $d=2$ | $d=3$ |
| NeurASP | $10.00_{\pm0.00}$ | $9.73_{\pm0.46}$ | $9.73_{\pm0.46}$ | N/A | $6.27_{\pm0.85}$ | N/A | N/A |
| DeepProbLog | $7.55_{\pm0.58}$ | N/A | N/A | N/A | $6.56_{\pm0.73}$ | N/A | N/A |
| DeepStochLog | $70.56_{\pm0.89}$ | $73.25_{\pm1.40}$ | $72.88_{\pm1.30}$ | $71.80_{\pm1.50}$ | $49.25_{\pm2.34}$ | $42.42_{\pm1.72}$ | $35.10_{\pm2.03}$ |
| LTN | $68.25_{\pm1.08}$ | $71.10_{\pm1.52}$ | $71.77_{\pm1.18}$ | $70.95_{\pm1.36}$ | $52.85_{\pm1.88}$ | $53.59_{\pm2.01}$ | $51.14_{\pm2.22}$ |
| ABL | $70.05_{\pm1.10}$ | $74.49_{\pm1.82}$ | $74.50_{\pm1.06}$ | $73.05_{\pm1.29}$ | $60.87_{\pm1.22}$ | $61.75_{\pm1.16}$ | $64.14_{\pm1.32}$ |
| A$^3$BL | $\mathbf{72.06}_{\pm\mathbf{1.12}}$ | $75.62_{\pm1.59}$ | $74.65_{\pm1.72}$ | $73.28_{\pm1.52}$ | $22.45_{\pm5.25}$ | $60.75_{\pm1.58}$ | $65.81_{\pm1.12}$ |
| **C-ABL** | $71.77_{\pm1.03}$ | $\mathbf{76.74}_{\pm\mathbf{1.34}}$ | $\mathbf{77.65}_{\pm\mathbf{1.32}}$ | $\mathbf{76.30}_{\pm\mathbf{1.22}}$ | $\mathbf{63.02}_{\pm\mathbf{1.03}}$ | $\mathbf{64.25}_{\pm\mathbf{1.12}}$ | $\mathbf{66.67}_{\pm\mathbf{1.11}}$ |
| NeurASP | $113.2_{\pm7.8}$ | $214.7_{\pm13.6}$ | $376.5_{\pm22.9}$ | N/A | $108.5_{\pm10.3}$ | N/A | N/A |
| DeepProbLog | $269.8_{\pm7.7}$ | N/A | N/A | N/A | $214.3_{\pm14.2}$ | N/A | N/A |
| DeepStochLog | $32.8_{\pm2.1}$ | $59.2_{\pm3.8}$ | $101.4_{\pm7.9}$ | $492.5_{\pm22.4}$ | $34.5_{\pm2.3}$ | $97.3_{\pm5.9}$ | $182.4_{\pm9.1}$ |
| LTN | $30.2_{\pm2.0}$ | $55.6_{\pm3.5}$ | $96.7_{\pm6.8}$ | $473.2_{\pm25.1}$ | $33.1_{\pm2.0}$ | $93.5_{\pm5.7}$ | $176.0_{\pm8.8}$ |
| ABL | $\mathbf{12.6}_{\pm\mathbf{1.1}}$ | $27.4_{\pm2.1}$ | $47.1_{\pm3.2}$ | $253.6_{\pm14.8}$ | $20.8_{\pm1.2}$ | $49.1_{\pm3.1}$ | $99.7_{\pm5.2}$ |
| A$^3$BL | $20.6_{\pm1.9}$ | $29.1_{\pm2.4}$ | $60.1_{\pm4.0}$ | $291.2_{\pm18.2}$ | $31.2_{\pm2.6}$ | $68.2_{\pm4.3}$ | $109.8_{\pm6.8}$ |
| **C-ABL** | $16.2_{\pm1.4}$ | $\mathbf{21.5}_{\pm\mathbf{1.9}}$ | $\mathbf{39.8}_{\pm\mathbf{2.8}}$ | $\mathbf{132.6}_{\pm\mathbf{7.7}}$ | $\mathbf{16.4}_{\pm\mathbf{1.9}}$ | $\mathbf{32.0}_{\pm\mathbf{2.0}}$ | $\mathbf{55.5}_{\pm\mathbf{4.2}}$ |

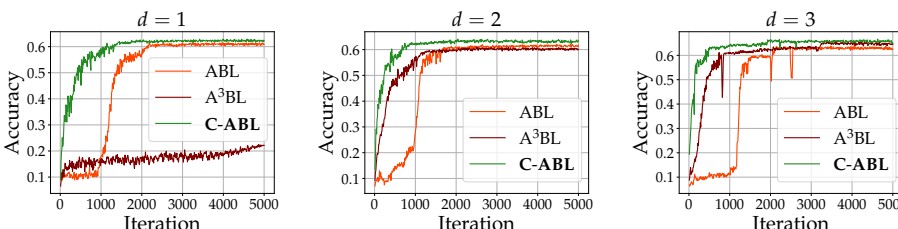

Figure 2: Comparison of training curves on the hexadecimal addition task with varying digit length $d$.

Experiments were conducted on decimal addition ($d = 1$ to 4) and hexadecimal addition ($d = 1$ to 3). As shown in Table 1, compared to previous ABL and other neuro-symbolic methods, C-ABL consistently achieves higher accuracy and requires significantly less total training time, indicating a more efficient reasoning process within each iteration. This improvement becomes especially evident in more complex settings (e.g., higher-digit or hexadecimal tasks), where the knowledge base is larger and reasoning becomes more expensive, highlighting the benefits of curriculum-style learning.

**Analysis of the Training Process.** We examine how curriculum learning affects ABL training process. Figure 2 shows the training curves on the hexadecimal setting with varied $d$. We may see that C-ABL achieves faster and more stable convergence than ABL and A$^3$BL across all settings: While the baselines often exhibit noisy or stagnant training curves, C-ABL rapidly improves within the first few hundred iterations and reaches near-maximum performance by iteration 1,000. This aligns with our theoretical findings: C-ABL mitigates training instability and achieves faster convergence.

Figure 3 further shows the training curves of three randomly selected concept labels in C-ABL (shown in green curves), each introduced in a different phase, with three vertical dashed lines indicating the start of each phase. (Here we present the first 2,000 iterations; the full training curves are provided in Figure 6 in Appendix F.1.) As shown, each concept improves immediately and stably once its corresponding phase begins, while learned concepts from earlier phases remain unaffected. In contrast, ABL introduces all labels at once (shown in red curves), leading to early-stage oscillations and slower convergence for all labels. This support our claim: C-ABL enables stable, stepwise learning with smooth transitions across curriculum phases.

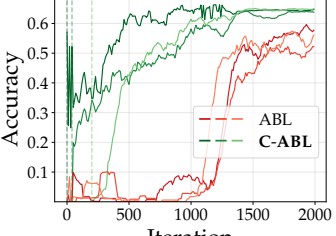

Figure 3: Training curve of three concept labels (hex., $d = 1$), each introduced in a different phase.

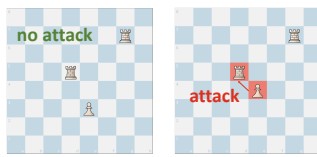

Figure 4: Examples of chess attack task.

Table 2: Results on chess attack task.

| Method | Accuracy ↑ | Iterations to Converge ↓ |
|---|---|---|
| ABL | $73.75_{\pm 0.84}$ | $4560_{\pm 96}$ |
| A³BL | $74.96_{\pm 0.78}$ | $3389_{\pm 105}$ |
| **C-ABL** | $\mathbf{86.79_{\pm 0.66}}$ | $\mathbf{2553_{\pm 88}}$ |

Table 3: Results on judicial sentencing task. C-ABL achieves improved F1, MAE and MSE scores, and takes fewer tokens to converge (defined as the point where F1 reaches 99% of its final value).

| Method | Labeled Data % | F1 ↑ | MAE ↓ | MSE ↓ | Tokens to Converge ↓ |
|---|---|---|---|---|---|
| ABL | 10% | $0.895_{\pm 0.024}$ | $\mathbf{0.762_{\pm 0.018}}$ | $0.888_{\pm 0.069}$ | 4.100M |
| **C-ABL** |  | $\mathbf{0.904_{\pm 0.011}}$ | $\mathbf{0.762_{\pm 0.036}}$ | $\mathbf{0.865_{\pm 0.094}}$ | **3.199M** |
| ABL | 50% | $0.910_{\pm 0.016}$ | $0.773_{\pm 0.010}$ | $0.874_{\pm 0.022}$ | 3.101M |
| **C-ABL** |  | $\mathbf{0.920_{\pm 0.006}}$ | $\mathbf{0.722_{\pm 0.014}}$ | $\mathbf{0.773_{\pm 0.048}}$ | **2.106M** |

### 5.2 Chess Attack

We consider a chess attack task [Dai et al., 2019, Huang et al., 2023]. The input is a randomly generated chessboard populated with chess pieces, each represented by an image, and the goal is to determine whether any pair of pieces are in an attacking relationship, see examples in Figure 4. We aim to assign a concept label to each piece from $\mathcal{Z} = \{\texttt{rook}, \texttt{pawn}, \texttt{bishop}, \texttt{king}, \texttt{knight}, \texttt{queen}\}$, such that the assignments can logically determine the boolean target label (attack). The knowledge base defines the attack behavior of each piece. In our setup, chess pieces are represented using randomly sampled MNIST digits, and we use LeNet-5 [LeCun et al., 1998] as the perception model. More details are provided in Appendix F.2.

We compare C-ABL with ABL and A³BL, with results shown in Table 2. C-ABL achieves higher accuracy with faster convergence. Notably, since the target label in this task is binary, the abduction space is particularly large, highlighting the advantage of C-ABL in complex reasoning scenarios.

### 5.3 Judicial Sentencing

In this section, we apply C-ABL to a real-world task: judicial sentencing. This task aims to predict the final sentence length $y \in \mathbb{R}^+$ based on the input criminal judgment records. The model first predicts concept labels $z$ representing sentencing factors (e.g., "voluntary surrender", "repeat offense"), and then reasons with a legal knowledge base $\mathcal{KB}$ to deduce $y$. We use the dataset from Huang et al. [2020], which includes 687 records. We use the pretrained `google-bert/bert-base-chinese` [Devlin et al., 2019] as the learning model. More details are provided in Appendix F.3.

We compare C-ABL with prior ABL implementations under two semi-supervised training settings, where 10% and 50% of the data include ground-truth concept labels available for training supervision. As shown in Table 3, C-ABL achieves higher F1 scores, lower MAE and MSE, and requires fewer tokens to converge, showing superior accuracy and training efficiency in real-world setting.

## 6 Related Work

Recently, there has been notable progress in combining data-driven machine learning with knowledge-driven logical reasoning. Some methods embed logical structure into neural networks [Serafini and Garcez, 2016, Xu et al., 2018, Badreddine et al., 2022, Yang et al., 2022, Ahmed et al., 2022], often by relaxing logical constraints, which can undermine reliability. Probabilistic neuro-symbolic systems preserve classical reasoning by treating neural outputs as distributions over symbols [Manhaeve et al., 2018, Yang et al., 2020, Kikaj et al., 2025]. Their inference typically involves weighted model counting [Chavira and Darwiche, 2008], whose exact computation can be intractable for large or complex knowledge bases. Recent efforts have introduced approximate inference methods to alleviate this issue [Manhaeve et al., 2021b, Winters et al., 2022, van Krieken et al., 2023, Li et al., 2023]. Abductive Learning (ABL) [Zhou, 2019, Zhou and Huang, 2022] integrates learning and reasoning in a balanced loop, where abduction and consistency optimization provide a bridge between the two

components. Supported by an open-source toolkit [Huang et al., 2024], ABL has shown success in diverse applications [Huang et al., 2020, Cai et al., 2021, Wang et al., 2021a, Gao et al., 2024, Hu et al., 2025, Wang et al., 2025]. However, current implementations treat the knowledge base as a black box and perform reasoning over the full logical space, often resulting in unstable supervision and inefficient convergence when the logical knowledge base is large or complex.

Curriculum learning improves generalization by organizing the learning process from easier to harder [Bengio et al., 2009, Wang et al., 2021b]. In the neuro-symbolic field, prior work has incorporated curriculum strategies by gradually increasing the complexity of input data or task structure [Mao et al., 2019, Chen et al., 2020, Le-Phuoc et al., 2021, Marconato et al., 2023a, Abbe et al., 2023, Lorello et al., 2024, Wei et al., 2025]. While they mainly focus on the data side, C-ABL introduces a curriculum over the knowledge base itself—leveraging its symbolic structure to guide reasoning progression. This allows the model to engage with increasingly complex logic in a controlled, interpretable way, and better align with the goal of exploiting structured knowledge.

Another line of research investigates ambiguity and shortcut behavior in neuro-symbolic systems and ABL framework [Marconato et al., 2023a, 2024, Li et al., 2024, Yang et al., 2024, He et al., 2024], which shares an underlying connection with our analysis: Large abduction spaces in ABL introduce ambiguity in concept label selection, which can in turn result in unstable or shortcut-driven supervision [He et al., 2024]. C-ABL tackles this structually by actively managing the reasoning space of knowledge base, thereby enhancing the reliability of reasoning supervision.

Partitioning knowledge bases into structured components has long been studied in the logic and knowledge representation communities [Amir and McIlraith, 2000, 2005, Gocht and Balyo, 2017, Simiński, 2017]. While originally developed for formal reasoning, these ideas motivate our logic-guided curriculum for improving learning dynamics in ABL.

## 7   Conclusion

This paper presents Curriculum Abductive Learning (C-ABL), a method designed to improve the training stability and efficiency of Abductive Learning (ABL). Unlike prior methods that treat the knowledge base as a fixed black box, C-ABL leverages structual properties of knowledge base, partitions the logic into sub-bases and introduces them progressively. This curriculum design reduces the abduction space, stabilizes supervision, and enables the model to incorporate domain knowledge from simple to complex. We provide theoretical guarantees for phase-level improvements and smooth transitions, and demonstrate strong empirical results on synthetic and real-world tasks. By structurally organizing domain knowledge, C-ABL offers a principled path toward scalable and efficient ABL.

## Acknowledgements

This research was supported by the Jiangsu Science Foundation Leading-edge Technology (BK20232003) and the National Natural Science Foundation of China (62176117). Cunjing Ge was supported by the National Natural Science Foundation of China (62202218). The authors would like to thank Reviewer #dk43 of OpenReview for the thorough review and insightful suggestions.

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

# A Limitation

While Curriculum Abductive Learning (C-ABL) demonstrates substantial improvements in training stability and reasoning performance, it also has several limitations. First, our partitioning strategy assumes that the knowledge base exhibits certain structural regularity—such as modularity or dependency chains—that supports meaningful curriculum design. This assumption holds for many real-world domains, including law, mathematics, medicine, structured game, etc. However, in some cases where the rule set is densely entangled or lacks clear structure, such partitioning may be less effective. Second, the current design of phase transitions is based on fixed structural principles and a simple accuracy threshold; more adaptive or learnable curriculum schedules could further improve performance but remain unexplored. Third, like standard ABL, C-ABL assumes access to a pre-defined, human-curated knowledge base. While our focus is on leveraging such structured knowledge, future work may explore methods for acquiring or refining the knowledge base, potentially by leveraging information from data.

# B Additional Analysis of Algorithm 1

## B.1 Examples of Knowledge Base Partitioning

This section illustrates examples of partitioning the knowledge base into sub-bases based on Algorithm 1 in Section 2.1.

**Chess Attack.** We use the chess attack task as the first example. As stated in Section 5.2, in this task, the input is a chessboard configuration with several pieces, and the concept label set is $\mathcal{Z} = \{\texttt{rook}, \texttt{pawn}, \texttt{bishop}, \texttt{king}, \texttt{knight}, \texttt{queen}\}$. $\mathcal{KB}$ contains rules of the attack behavior for each piece, with the target label $y = \texttt{attack}$, indicating whether any pair of pieces in the chessboard are in an attacking relation.

Figure 5 illustrates a portion the aforementioned $\mathcal{KB}$, and provides a visual example of how the dependency graph is constructed as well as how sub-base clusters are formed based on Algorithm 1.

Each node in the dependency graph corresponds to a rule in the knowledge base, and a directed edge is drawn from rule $r_i$ to $r_j$ if the head predicate of $r_i$ appears in the body of $r_j$. For example, in the top portion of the figure, the rule for $\texttt{lshape}$ depends on both $\texttt{left}$ and $\texttt{fwd}$, and in turn supports the rule for $\texttt{attack}$ when the piece is a $\texttt{knight}$. This forms a dependency path: $\texttt{left}, \texttt{fwd} \rightarrow \texttt{lshape} \rightarrow \texttt{attack}$.

Our algorithm begins by constructing an initial cluster $C_z$ for each concept label $z \in \mathcal{Z}$ by collecting all rules that directly reference $z$ in their bodies. In the figure, these clusters are color-coded by background:

- The **green-shaded region** corresponds to $C_{\texttt{knight}}$, which includes the $\texttt{attack}$ rule with concept $\texttt{knight}$, the $\texttt{lshape}$ rule, and the supporting geometric rules $\texttt{left}$ and $\texttt{fwd}$.

- The **red-shaded region** represents $C_{\texttt{rook}}$, which includes the $\texttt{attack}$ rule with concept $\texttt{rook}$, and the supporting $\texttt{line}$ predicate rules that it depends on.

- The **yellow-shaded region** represents $C_{\texttt{bishop}}$, which includes the $\texttt{attack}$ rule with concept $\texttt{bishop}$, and the supporting $\texttt{diag}$ predicate rules that it depends on.

- The **blue-shaded region** shows $C_{\texttt{queen}}$, which includes the $\texttt{attack}$ rule involving $\texttt{queen}$ and the $\texttt{line\_or\_diag}$ predicate, along with its dependencies on both $\texttt{line}$ and $\texttt{diag}$.

Edges in the figure indicate predicate-level dependencies between rules, which guide the recursive expansion of clusters and the establishment of inter-cluster precedence. For example, $C_{\texttt{queen}}$ depends on both $C_{\texttt{rook}}$ and $C_{\texttt{bishop}}$ due to its reliance on $\texttt{line}$ and $\texttt{diag}$ via $\texttt{line\_or\_diag}$, so both $C_{\texttt{rook}}$ and $C_{\texttt{bishop}}$ must precede $C_{\texttt{queen}}$ in the curriculum.

Figure 5: A portion of knowledge base in chess attack task.

The resulting sub-bases, for example, follow an incremental structure such as:

$$\begin{aligned}
\mathcal{KB}_1 &= C_{\texttt{knight}} \\
\mathcal{KB}_2 &= C_{\texttt{knight}} \cup C_{\texttt{rook}} \\
\mathcal{KB}_3 &= C_{\texttt{knight}} \cup C_{\texttt{rook}} \cup C_{\texttt{bishop}} \\
\mathcal{KB}_4 &= C_{\texttt{knight}} \cup C_{\texttt{rook}} \cup C_{\texttt{bishop}} \cup C_{\texttt{queen}}
\end{aligned} \qquad (2)$$

$d$-**digit Addition.** In the digit addition task (outlined in Example 2.1 and Section 5.1), the knowledge base $\mathcal{KB}$ encodes multi-digit addition through symbolic predicates for digits and arithmetic operations. It includes logic for digit-wise computation and mappings from concept labels (e.g., zero, one) to numeric values, as illustrated below:

$$\begin{aligned}
\texttt{addition}(\texttt{Num1}, \texttt{Num2}, \texttt{Y}) &\leftarrow \texttt{number}(\texttt{Num1}, \texttt{Res1}), \\
&\quad \texttt{number}(\texttt{Num2}, \texttt{Res2}), \\
&\quad \texttt{Y is Res1} + \texttt{Res2}. \\
\texttt{number}([], \texttt{Res}, \texttt{Res}) &\leftarrow. \\
\texttt{number}([\texttt{H}|\texttt{T}], \texttt{Acc}, \texttt{Res}) &\leftarrow \texttt{digit}(\texttt{H}, \texttt{D}), \\
&\quad \texttt{Acc1 is D} + 10 * \texttt{Acc}, \\
&\quad \texttt{number}(\texttt{T}, \texttt{Acc1}, \texttt{Res}). \\
\texttt{number}(\texttt{X}, \texttt{N}) &\leftarrow \texttt{number}(\texttt{X}, 0, \texttt{N}). \\
\texttt{digit}(\texttt{Pos}, 0) &\leftarrow \texttt{zero}(\texttt{Pos}). \\
\texttt{digit}(\texttt{Pos}, 1) &\leftarrow \texttt{one}(\texttt{Pos}). \\
\texttt{digit}(\texttt{Pos}, 2) &\leftarrow \texttt{two}(\texttt{Pos}). \\
\texttt{digit}(\texttt{Pos}, 3) &\leftarrow \texttt{three}(\texttt{Pos}). \\
\texttt{digit}(\texttt{Pos}, 4) &\leftarrow \texttt{four}(\texttt{Pos}). \\
\texttt{digit}(\texttt{Pos}, 5) &\leftarrow \texttt{five}(\texttt{Pos}). \\
\texttt{digit}(\texttt{Pos}, 6) &\leftarrow \texttt{six}(\texttt{Pos}). \\
\texttt{digit}(\texttt{Pos}, 7) &\leftarrow \texttt{seven}(\texttt{Pos}). \\
\texttt{digit}(\texttt{Pos}, 8) &\leftarrow \texttt{eight}(\texttt{Pos}). \\
\texttt{digit}(\texttt{Pos}, 9) &\leftarrow \texttt{nine}(\texttt{Pos}).
\end{aligned}$$

When applying Algorithm 1, rules are partitioned based on concept label dependencies. For example, $\mathcal{KB}_1$ may include rules for $\mathcal{Z}_1 = \{\mathtt{zero}, \mathtt{one}\}$, then expand to $\mathcal{KB}_2$ with $\mathcal{Z}_2 = \{\mathtt{zero}, \mathtt{one}, \mathtt{two}, \mathtt{three}\}$, and so on. Each $\mathcal{KB}_p$ includes (1) digit-mapping rules relevant to $\mathcal{Z}_p$, and (2) shared arithmetic rules for addition (rules with predicate $\mathtt{addition}$ as head) and number construction (rules with predicate $\mathtt{number}$ as head). While the core arithmetic logic remains fixed across phases, digit-related rules are introduced gradually.

The above two visualizations on chess attack and digit addition make clear how our algorithm traces dependencies to group semantically related rules into coherent sub-bases, and defines a curriculum order based on the reasoning hierarchy encoded in the knowledge base structure.

## B.2 Computational Cost Analysis

We now analyze the computational overhead introduced by the knowledge base partitioning algorithm, i.e., Algorithm 1 proposed in Section 3.1.

**Time Complexity.** Let $n$ and $e$ be the number of nodes (also, number of rules in $\mathcal{KB}$) and edges in the dependency graph of $\mathcal{KB}$. The partitioning process involves the following major steps:

1. **Dependency Graph Construction:** For each rule, we inspect its head and compare it against all other rule bodies. This results in $\mathcal{O}(n^2)$ comparisons in the worst case. Also, since predicates tend to be sparse in practice, a hash-based indexing scheme can reduce this to $\mathcal{O}(n + e)$.

2. **Cluster Initialization and Expansion:** For each concept label $z \in \mathcal{Z}$ (with $|\mathcal{Z}| = N$), we initiate a cluster and expand it by traversing the dependency graph, the overall time complexity of this process is $\mathcal{O}(N(n + e))$.

3. **Precedence Ordering and Topological Sort:** After forming clusters, we identify precedence relations based on cross-cluster dependencies. This forms a directed acyclic graph over clusters, with topological sorting in $\mathcal{O}(N + e_c)$ time, where $e_c$ is the number of inter-cluster edges (typically $e_c \ll e$).

Putting all steps together, the overall complexity is approximately $\mathcal{O}(N(n + e))$ under practical sparsity assumptions.

**Empirical Partition Time.** In practice, we evaluated the partitioning time on the two examples described in Appendix B.1. For the digit addition task, the knowledge base contains 14 rules, and partitioning takes 0.05 seconds. For the chess attack task, which includes 58 rules, partitioning completes in 0.9 seconds. All measurements were conducted on a standard CPU (Intel Xeon Gold 6226R, single thread), with no GPU required.

To put this in context, a full ABL training run (e.g., under 1,000 iterations) typically takes several minutes to hours, depending on the task and model. Therefore, the knowledge base partitioning accounts for a minimal portion of the total runtime. More importantly, it is a **one-time, offline preprocessing** step, independent of data size or iteration count, and its cost amortizes over training. In summary, Algorithm 1 incurs negligible computational overhead while providing benefits in reasoning efficiency, supervision quality, and convergence speed.

# C Theorem Proof

## C.1 Proof of Theorem 3.2

*Proof.* We prove the two directions separately.

**Soundness.** This follows from the fact that $\mathcal{KB}_p \subseteq \mathcal{KB}$, Any logical derivation that holds under $\mathcal{KB}_p$ also holds under the full knowledge base $\mathcal{KB}$, since $\mathcal{KB}$ contains all rules in $\mathcal{KB}_p$. Therefore, we have:

$$\mathcal{KB}_p \models \varphi \implies \mathcal{KB} \models \varphi. \tag{3}$$

**Completeness.** We assume that $\mathcal{KB} \models \varphi$, i.e., there exists a proof (derivation) of $\varphi$ using rules from $\mathcal{KB}$. Let $\mathcal{R} \subseteq \mathcal{KB}$ denote the minimal set of rules used in this derivation. We now argue that all rules in $\mathcal{R}$ are also contained in $\mathcal{KB}_p$.

As stated in Algorithm 1, for any predicate $z \in \mathcal{Z}_p$, sub-base involving it is constructed by: (1) Create a cluster $C_z$ by collecting all rules whose bodies mention $z$. (2) Recursively expand $C_z$ by traversing the dependency graph $G$: If a rule depends on a predicate $a$, and $a$ is concluded by another rule, then that rule is also included. (3) Continue until all rules needed to derive any conclusions involving $z$ are included. Therefore, for each $z \in \mathcal{Z}_p$, all reasoning chains involving $z$ and the predicates it depends on are included in $\mathcal{KB}_p$ by construction. Since by assumption, $\varphi$ only involves predicates in $\mathcal{Z}_p$, and the derivation of $\varphi$ in $\mathcal{KB}$ uses only predicates and rules in these chains, we conclude that $\mathcal{R} \subseteq \mathcal{KB}_p$.

Hence, the same inference of $\varphi$ can be carried out within $\mathcal{KB}_p$. Therefore, we have:

$$\mathcal{KB} \models \varphi \implies \mathcal{KB}_p \models \varphi. \tag{4}$$

$\square$

## C.2 Proof of Corollary 3.3

*Proof.* Since the final phase $P$ includes all clusters and thus covers all predicates in $\mathcal{Z}$, we have $\mathcal{Z}_P = \mathcal{Z}$. Therefore, for any formula $\varphi$ over $\mathcal{Z}$, we trivially have:

$$\mathcal{KB}_P \models \varphi \iff \mathcal{KB} \models \varphi. \tag{5}$$

$\square$

## C.3 Proof of Lemma 4.1

*Proof.* Each of the $m$ positions independently selects a label from $\mathcal{Z}$, giving $N^m$ possible assignments in total. Logical constraints may eliminate some assignments, but the maximum remains $N^m$. $\square$

## C.4 Proof of Theorem 4.2

*Proof.* We first denote $\mathcal{Z}_p = \mathcal{Z}_{p-1} \cup \Delta\mathcal{Z}_p$, where $\Delta\mathcal{Z}_p = \mathcal{Z}_p \setminus \mathcal{Z}_{p-1}$ are the new concept labels introduced in phase $p$, and for any $\boldsymbol{z} \in \mathbb{S}_p$, we write $\boldsymbol{z} = (\boldsymbol{z}_{\text{old}}, \boldsymbol{z}_{\text{new}})$ accordingly.

Recall that the consistency score used for pseudo-label selection is:

$$\text{Con}(\boldsymbol{z}, f) = \prod_{i=1}^{m} p^{(i)}(z^{(i)}). \tag{6}$$

where $p^{(i)}(z^{(i)})$ denotes the model's predicted probability for concept $z^{(i)}$ at position $i$.

Now fix two candidate concept sequences $\boldsymbol{z}^{(a)}$ and $\boldsymbol{z}^{(b)}$ in $\mathbb{S}_p$, such that: $\boldsymbol{z}^{(a)}$ and $\boldsymbol{z}^{(b)}$ agree on all positions $i$ where $z^{(i)} \in \mathcal{Z}_p \setminus \mathcal{Z}_{p-1}$, but disagree at some positions $i$ where $z^{(i)} \in \mathcal{Z}_{p-1}$. Then their consistency scores differ only in terms involving $\mathcal{Z}_{p-1}$. By the semantic stability assumption, previously learned concepts retain their meaning across phases. Combined with the assumption that for any previously introduced concept $z \in \mathcal{Z}_{p-1}$, the model achieves accuracy exceeding random chance, so that for positions $i$ with $z^{(i)} \in \mathcal{Z}_{p-1}$, the predicted probabilities $p^{(i)}(z^{(i)})$ tend to assign higher mass to correct concepts learned in prior phases. Therefore we have $\text{Con}(\boldsymbol{z}^{(a)}, f) > \text{Con}(\boldsymbol{z}^{(b)}, f)$, meaning the selection strategy will typically prefer $\boldsymbol{z}^{(a)}$ over $\boldsymbol{z}^{(b)}$.

This shows that, under consistency-based selection, the model tends to preserve earlier learned concepts (those in $\mathcal{Z}_{p-1}$), and variations in $\mathbb{S}_p$ primarily arise from the newly introduced concepts $\mathcal{Z}_p \setminus \mathcal{Z}_{p-1}$. Therefore, the number of valid configurations in $\mathbb{S}_p$ is bounded by the number of possible combinations of $\boldsymbol{z}_{\text{new}}$:

$$|\mathbb{S}_p| \leq |\Delta\mathcal{Z}_p|^m = |\mathcal{Z}_p \setminus \mathcal{Z}_{p-1}|^m. \tag{7}$$

$\square$

## C.5 Proof of Theorem 4.5

*Proof.* Let the loss function at iteration $t$ be defined as:

$$L^{(t)} = \ell(f^{(t)}(x), \bar{\boldsymbol{z}}^{(t)}), \tag{8}$$

where $\bar{\boldsymbol{z}}^{(t)} \in \mathbb{S}$ is selected via abduction given $y$ and $\mathcal{KB}$, we aim to bound the average suboptimality of the iterates:

$$\frac{1}{T} \sum_{t=1}^{T} \mathbb{E}[L^{(t)} - L^*] \leq \varepsilon, \tag{9}$$

where $L^* = \min_f \ell(f(x), \boldsymbol{z}^*)$ is the best possible loss under the true concept label $\boldsymbol{z}^*$.

By chapter 14 in Shalev-Shwartz and Ben-David [2014], under the assumptions that: (1) loss function $L^{(t)}$ is convex, (2) the gradient norm is bounded by the Lipschitz constant $\rho$, i.e., $\|\nabla L^{(t)}\| \leq \rho$, and (3) the model lies in a bounded domain of diameter $D$, then the regret over $T$ steps is bounded by:

$$\sum_{t=1}^{T} L^{(t)} - L^* \leq \frac{D^2}{2\eta} + \frac{\eta T \rho^2}{2}. \tag{10}$$

Choosing the optimal learning rate $\eta = \frac{D}{\rho\sqrt{T}}$, we obtain:

$$\frac{1}{T} \sum_{t=1}^{T} \mathbb{E}\left[L^{(t)} - L^*\right] \leq \frac{D\rho}{\sqrt{T}}. \tag{11}$$

To ensure this average suboptimality is below $\varepsilon$, it suffices that:

$$\frac{D\rho}{\sqrt{T}} \leq \varepsilon \implies T \geq \left(\frac{D\rho}{\varepsilon}\right)^2. \tag{12}$$

Now, suppose that due to the abductive process, at each iteration $t$, the pseudo-label $\bar{\boldsymbol{z}}^{(t)}$ is selected uniformly from a candidate set of size $|\mathbb{S}|$, then the effective variance or inconsistency in supervision may grow linearly with $|\mathbb{S}|$. To conservatively account for this, we assume the domain diameter $D = \mathcal{O}(|\mathbb{S}|)$, which reflects the number of pseudo-label variants over which the model needs to generalize. Therefore, we have

$$T = \mathcal{O}\left(\frac{|\mathbb{S}|^2 \cdot \rho^2}{\varepsilon^2}\right). \tag{13}$$

which completes the proof. $\square$

## C.6 Proof of Theorem 4.6

*Proof.* Recall that for each phase $p$, we have $\mathcal{KB}_p \subseteq \mathcal{KB}_{p+1}$ and that $\mathcal{Z}_p \subseteq \mathcal{Z}_{p+1}$, therefore, by Theorem 3.2, both $\mathcal{KB}_p$ and $\mathcal{KB}_{p+1}$ are sound and complete with respect to all formulas over $\mathcal{Z}_p$. Then, for any formula $\varphi$ over $\mathcal{Z}_p$, we have:

$$\mathcal{KB}_p \models \varphi \iff \mathcal{KB} \models \varphi \iff \mathcal{KB}_{p+1} \models \varphi \tag{14}$$

where the first equivalence comes from applying Theorem 3.2 to $\mathcal{KB}_p$ and the second from applying it to $\mathcal{KB}_{p+1}$, since both sub-bases fully capture all reasoning over the same concept domain $\mathcal{Z}_p$.

Hence, $\mathcal{KB}_p \models \varphi$ if and only if $\mathcal{KB}_{p+1} \models \varphi$, completing the proof. $\square$

## C.7 Background on Stone Spaces and Proof of Theorem 4.7

We first provide backgrounds on stone spaces [Johnstone, 1982]: Stone spaces provide a topological perspective on logic by characterizing the space of all models (truth assignments) that satisfy a given set of logical formulas. Formally, given a Boolean algebra $\mathcal{B}$, its Stone space $S(\mathcal{B})$ is defined as the

set of all ultrafilters over $\mathcal{B}$, where an ultrafilter is a maximal consistent set of formulas—intuitively, it corresponds to a complete and consistent truth assignment over the logic.

Stone's representation theorem [Stone, 1936] states that every Boolean algebra is isomorphic to the algebra of clopen (simultaneously closed and open) subsets of its Stone space. This construction enables logical entailment to be interpreted as topological containment: if $\varphi \in \mathcal{B}$ holds in all ultrafilters in $S(\mathcal{B})$, then $\varphi$ is a logical consequence of $\mathcal{B}$.

In our setting, for each phase $p$, we define a Boolean algebra $\mathcal{B}_p$ as the closure of all logical formulas over the predicate set $\mathcal{Z}_p$. The Stone space $S(\mathcal{B}_p)$ thus encodes all logically consistent concept-level assignments admissible under $\mathcal{KB}_p$.

Below is the proof of Theorem 4.7.

*Proof.* Recall that for each phase $p$, we construct a sub-base $\mathcal{KB}_p$ such that $\mathcal{KB}_p \subseteq \mathcal{KB}_{p+1}$ and that $\mathcal{Z}_p \subseteq \mathcal{Z}_{p+1}$. Let $\mathcal{B}_p$ denote the Boolean algebra of formulas over $\mathcal{Z}_p$, and $\mathcal{B}_{p+1}$ denote that over $\mathcal{Z}_{p+1}$. Since $\mathcal{Z}_p \subseteq \mathcal{Z}_{p+1}$, every formula over $\mathcal{Z}_p$ is also a formula over $\mathcal{Z}_{p+1}$, that is, the Boolean algebras satisfies $\mathcal{B}_p \subseteq \mathcal{B}_{p+1}$.

We now argue that any ultrafilter over $\mathcal{B}_p$ can be extended to an ultrafilter over $\mathcal{B}_{p+1}$: Following Zorn's Lemma [Conrad, 2016], any consistent set of formulas over a Boolean algebra can be extended to an ultrafilter over a larger Boolean algebra containing it, therefore, any model over $\mathcal{Z}_p$ can be extended to a model over $\mathcal{Z}_{p+1}$ that preserves the truth values of formulas in $\mathcal{B}_p$. Then, for every ultrafilter $U_p \in S(\mathcal{B}_p)$, there exists at least one ultrafilter $U_{p+1} \in S(\mathcal{B}_{p+1})$ such that $U_{p+1} \cap \mathcal{B}_p = U_p$. This directly implies:

$$S(\mathcal{B}_p) \subseteq S(\mathcal{B}_{p+1}), \tag{15}$$

since every model in phase $p$ remains valid (or extendable) in phase $p+1$.

$\square$

## D  Pseudocode for Curriculum Abductive Learning

This section shows the pseudocode (see Algorithm 2) for the Curriculum Abductive Learning (C-ABL) training process, as stated in Section 3.2.

---
**Algorithm 2:** Curriculum Abductive Learning (C-ABL) Training

---
**Input:** Sub-bases $\mathcal{KB}_1, \ldots, \mathcal{KB}_P$, training dataset $\mathcal{D}$
**Output:** Trained model $f$
1   Initialize $f$ randomly;
2   **for** $p = 1$ **to** $P$ **do**
3      Schedule training stream $\mathcal{D}_p \subseteq \mathcal{D}$ with concept labels in $\mathcal{Z}_p$;
4      **repeat**
5         **for** *each $x \in \mathcal{D}_p$* **do**
6            $\hat{z} \leftarrow f(x)$;
7            **if** $\hat{z} \wedge \mathcal{KB}_p \models y$ **then**
8               $\bar{z} \leftarrow \hat{z}$;
9            **else**
10               $\mathbb{S} \leftarrow \{z \mid z \wedge \mathcal{KB}_p \models y\}$;
11               $\bar{z} \leftarrow \underset{z \in \mathbb{S}}{\arg\min}\, 1 - \prod_i p^{(i)}(z^{(i)})$         $\triangleright$ from Eq. (1);
12            Update $f$ using $\bar{z}$;
13      **until** Phase transition condition met $(p < P)$ **or** termination condition met $(p = P)$ ;

---

## E  Comparison Methods

In this section, we provide a brief supplementary introduction to the compared baseline methods used in experiments.

**ABL and its Variants.**

(1) **ABL** [Dai et al., 2019], the original implementation of Abductive Learning, supported by an efficient open-source toolkit [Huang et al., 2024]. For consistency evaluation, we adopt the confidence-based selection strategy as defined in Section 2.3 (see Eq. (1));

(2) **A$^3$BL** [He et al., 2024], one of the most effective ABL variants, which improves concept label selection by evaluating all candidates in the abduction space.

Other variants such as Huang et al. [2021], Tao et al. [2024] also focus on refining label selection but perform less robustly than A$^3$BL. We do not compare with methods like ABL-Refl [Hu et al., 2025] or ABL-PSP [Jia et al., 2025], as they process the entire input sequence jointly and hence require different perception modules, making direct comparison less meaningful.

**Neuro-Symbolic Methods.**

(1) **NeurASP** [Yang et al., 2020], an extension of answer set programs [Brewka et al., 2011] by treating the neural network output as the probability distribution over atomic facts;

(2) **LTN** [Serafini and Garcez, 2016], a neural-symbolic framework that uses differentiable first-order logic language. This is a representative work of relaxing logical rules as soft constraints in neural networks;

(3) **DeepProbLog** [Manhaeve et al., 2018], an extension of ProbLog [De Raedt et al., 2007] by introducing neural predicates;

(4) **DeepStochLog** [Winters et al., 2022], a related strategy to DeepProbLog that improves inference efficiency through stochastic logic program.

# F   Experiment Details and Additional Results

All experiments are performed on a server with Intel Xeon Gold 6226R CPU and Tesla A100 GPU, and each experiment is repeated 5 times.

## F.1   $d$-digit Addition Tasks

**Curriculum Design.**   The curriculum partitioning of $\mathcal{KB}$ for this task is detailed in Appendix B.1. We set the minimum phase size to $\tau = 2$.

Additionally, Table 4 presents a sensitivity analysis on different values of $\tau$, conducted under the hexadecimal addition setting with $d = 3$. In this configuration, there are a total of 16 concept labels, and we experimented with $\tau$ values ranging from 1 to 4. Results show that our method consistently outperforms both ABL and A$^3$BL across different $\tau$ settings. In general, a smaller $\tau$ leads to finer-grained sub-bases and a smaller abduction space, which often improve reasoning accuracy and training efficiency. Indeed, when the phases are too fine-grained, the model will go through more curriculum, which slightly brings implementation complexity.

**Cross-Dataset Analysis.**   We further validate our method on alternative datasets. In the decimal setting, we now use digit images from MNIST dataset [Krizhevsky et al., 2009], and in the hexadecimal setting, we use digits 0–9 and letters A–F from EMNIST dataset [Cohen et al., 2017]. All methods use LeNet-5 [LeCun et al., 1998] as the perception model, and are trained for a total of 1,000 iterations. Results in Table 5 show that C-ABL continues to yield superior reasoning accuracy and training efficiency compared to baselines.

**Full Training Curve (Complementing Figure 3).**   Previously in Figure 3, we illustrated a segment of the training curves for three randomly selected concept labels, each introduced in a different phase (with three vertical dashed lines indicating the start of their respective phases). Here in Figure 6, we present the full training curve of these concept labels (left). We also provide a zoomed-in view (right) focusing on the early training period of C-ABL, which highlights that in C-ABL, each concept label improves immediately and stably upon the start of its corresponding curriculum phase, while the performance of previously learned concepts remains unaffected—demonstrating the smoothness and stability of phase transitions.

Table 4: Sensitivity analysis on different $\tau$ values across different tasks. C-ABL outperforms ABL and A$^3$BL across different $\tau$ settings.

| Method | | Digit Addition | | Chess Attack | |
|---|---|---|---|---|---|
| | | Test Accuracy | Training Time (min) | Test Accuracy | Training Time (min) |
| ABL | | $64.14_{\pm1.32}$ | $99.7_{\pm5.2}$ | $73.75_{\pm0.84}$ | $36.7_{\pm2.8}$ |
| A$^3$BL | | $65.81_{\pm1.12}$ | $109.8_{\pm6.8}$ | $74.96_{\pm0.78}$ | $45.6_{\pm3.2}$ |
| C-ABL | $\tau=1$ | $66.78_{\pm1.05}$ | $53.7_{\pm2.7}$ | $85.86_{\pm0.74}$ | $20.9_{\pm1.6}$ |
| | $\tau=2$ | $66.67_{\pm1.11}$ | $55.5_{\pm4.2}$ | $86.79_{\pm0.66}$ | $23.1_{\pm1.8}$ |
| | $\tau=3$ | $65.76_{\pm1.08}$ | $64.8_{\pm4.9}$ | $84.69_{\pm0.71}$ | $26.3_{\pm2.1}$ |
| | $\tau=4$ | $66.01_{\pm1.16}$ | $70.1_{\pm5.1}$ | – | – |

Table 5: Comparison of prediction accuracy (top) and total training time in minutes (bottom) on digit addition tasks with MNIST / EMNIST dataset. "N/A" indicates runtime exceeding 3 hours. C-ABL consistently achieves higher accuracy and faster training.

| Method | Decimal | | | | Hexadecimal | | |
|---|---|---|---|---|---|---|---|
| | $d=1$ | $d=2$ | $d=3$ | $d=4$ | $d=1$ | $d=2$ | $d=3$ |
| NeurASP | $97.01_{\pm0.24}$ | $10.21_{\pm0.39}$ | N/A | N/A | $77.65_{\pm6.29}$ | N/A | N/A |
| DeepProbLog | $97.42_{\pm0.29}$ | N/A | N/A | N/A | $96.59_{\pm0.48}$ | N/A | N/A |
| DeepStochLog | $99.00_{\pm0.13}$ | $98.72_{\pm0.14}$ | $98.59_{\pm0.12}$ | $98.46_{\pm0.11}$ | $97.56_{\pm0.12}$ | $97.96_{\pm0.11}$ | $97.13_{\pm0.15}$ |
| LTN | $98.64_{\pm0.18}$ | $98.53_{\pm0.21}$ | $98.48_{\pm0.17}$ | $98.34_{\pm0.19}$ | $97.18_{\pm0.14}$ | $96.53_{\pm0.17}$ | $97.11_{\pm0.13}$ |
| ABL | $98.79_{\pm0.10}$ | $98.99_{\pm0.08}$ | $98.59_{\pm0.09}$ | $98.29_{\pm0.11}$ | $97.17_{\pm0.10}$ | $97.49_{\pm0.08}$ | $97.50_{\pm0.06}$ |
| A$^3$BL | $98.94_{\pm0.12}$ | $99.03_{\pm0.09}$ | $98.68_{\pm0.10}$ | $98.42_{\pm0.12}$ | $96.76_{\pm0.12}$ | $97.62_{\pm0.09}$ | $97.65_{\pm0.07}$ |
| **C-ABL** | $\mathbf{99.04_{\pm0.06}}$ | $\mathbf{99.17_{\pm0.05}}$ | $\mathbf{99.09_{\pm0.04}}$ | $\mathbf{99.17_{\pm0.03}}$ | $\mathbf{98.48_{\pm0.03}}$ | $\mathbf{98.41_{\pm0.04}}$ | $\mathbf{98.83_{\pm0.02}}$ |
| NeurASP | $33.1_{\pm1.6}$ | $87.8_{\pm2.2}$ | N/A | N/A | $48.3_{\pm4.6}$ | N/A | N/A |
| DeepProbLog | $44.8_{\pm1.5}$ | N/A | N/A | N/A | $69.8_{\pm7.7}$ | N/A | N/A |
| DeepStochLog | $3.9_{\pm0.5}$ | $7.6_{\pm0.9}$ | $38.5_{\pm1.4}$ | $166.8_{\pm6.3}$ | $6.8_{\pm0.5}$ | $24.0_{\pm2.0}$ | $139.4_{\pm9.1}$ |
| LTN | $4.2_{\pm0.4}$ | $8.0_{\pm1.0}$ | $29.1_{\pm1.6}$ | $140.2_{\pm5.9}$ | $7.1_{\pm0.6}$ | $24.5_{\pm2.2}$ | $155.0_{\pm13.3}$ |
| ABL | $1.9_{\pm0.2}$ | $2.5_{\pm0.3}$ | $11.9_{\pm1.2}$ | $80.3_{\pm7.8}$ | $1.4_{\pm0.2}$ | $4.8_{\pm0.7}$ | $30.8_{\pm2.4}$ |
| A$^3$BL | $2.9_{\pm0.3}$ | $4.6_{\pm0.5}$ | $15.5_{\pm1.7}$ | $105.8_{\pm6.5}$ | $3.2_{\pm0.1}$ | $6.5_{\pm0.6}$ | $54.2_{\pm3.7}$ |
| **C-ABL** | $\mathbf{0.9_{\pm0.1}}$ | $\mathbf{1.8_{\pm0.2}}$ | $\mathbf{6.9_{\pm0.7}}$ | $\mathbf{33.2_{\pm2.3}}$ | $\mathbf{1.0_{\pm0.1}}$ | $\mathbf{3.4_{\pm0.4}}$ | $\mathbf{20.6_{\pm2.1}}$ |

## F.2 Chess Attack Task

**Curriculum Design.** The partitioning strategy for this task is also described in Appendix B.1. We again set the minimum phase size to $\tau=2$. A sensitivity analysis of this hyperparameter is provided also in Table 4, where we evaluate $\tau$ values ranging from 1 to 3 under the setting with 6 concept labels. Results show performance advantage of C-ABL over ABL and A$^3$BL in almost all tested $\tau$ values. As in the digit addition task, a smaller $\tau$ introduces more fine-grained sub-bases and smaller abduction spaces, which may lead to improved accuracy or faster convergence.

## F.3 Judicial Sentencing Tasks

**Dataset Description.** The dataset consists of 687 criminal judgment records for theft cases authored by judges in Guizhou, China, between 2017 and 2018 [Huang et al., 2020]. Each record includes detailed descriptions of the defendant's background and criminal record, as well as some facts and legal basis of the ruling.

**Knowledge Base.** The legal domain offers rich structured prior knowledge that can be encoded as first-order logic rules, forming the knowledge base $\mathcal{KB}$. These include statutory law, attribute-matching rules, and general common sense constraints. Below is an example adapted from Huang et al. [2020], which outlines how various attributes and the stolen amount of money determine the final sentence:

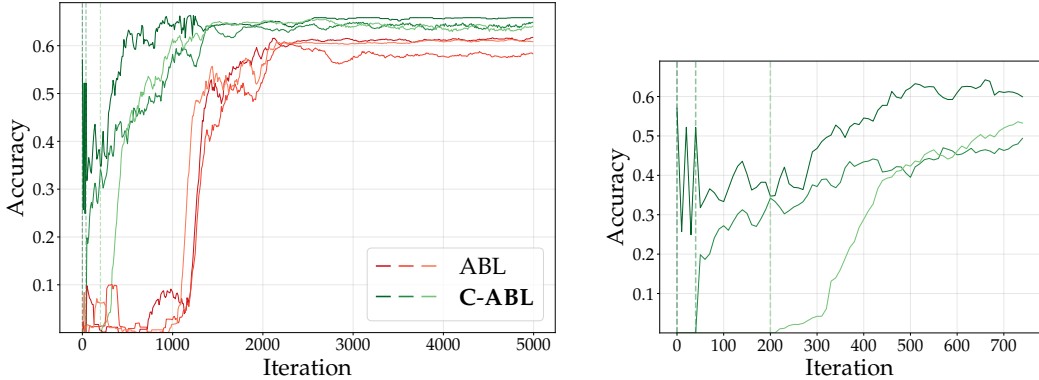

Figure 6: Full training curves of three concept labels introduced in different phases (left) and a zommed-in early-stage view for C-ABL (right).

$$
\begin{aligned}
\texttt{penalty}(\texttt{X}, \texttt{Y}) &\leftarrow \texttt{base\_penalty}(\texttt{X}, \texttt{Z}_1) \wedge \texttt{weight}(\texttt{X}, \texttt{Z}_2) \wedge \texttt{Y} = \texttt{Z}_1(1 + \texttt{Z}_2). \\
\texttt{base\_penalty}(\texttt{X}, \texttt{Y}) &\leftarrow \texttt{money}(\texttt{X}, m) \wedge \texttt{Y} = 0.7m + 5.7. \\
\texttt{weight}([], 0) &\leftarrow . \\
\texttt{weight}([\texttt{X}|\texttt{Xs}], \texttt{Y}) &\leftarrow \texttt{element\_weight}(\texttt{X}, \texttt{Z}_1) \wedge \texttt{weight}(\texttt{Xs}, \texttt{Z}_2) \wedge \texttt{Y} = \texttt{Z}_1 + \texttt{Z}_2.
\end{aligned}
$$

However, such symbolic knowledge is often incomplete—for instance, the rule "voluntary surrender reduces sentence" may exist, but the precise reduction is unspecified. To incorporate these partially defined patterns into a learnable system, we parametrize the knowledge base as:

$$
y = (1 + w^\top z)(am + b), \tag{16}
$$

where $z$ is a binary vector representing predicted case attributes (i.e., concept labels), $m \in \mathbb{N}$ is the amount of money involved (a ground-truth numeric input), and $y \in \mathbb{R}^+$ is the predicted sentence length. This formulation mirrors the logical structure of first-order rules while allowing the unknown parameters $w$, $a$, and $b$ to be learned via a linear regression model. Like the original rule-based system, Eq. (16) supports abduction: given a known sentence length $y$ and monetary amount $m$, the model can infer plausible case attributes $z$.

In summary, this task requires both the prediction of concept labels and the joint optimization of symbolic parameters to produce sentence predictions directly from judgment documents. The integration of symbolic reasoning and statistical learning makes this possible in a unified framework.

**Curriculum Design.** In C-ABL, the knowledge base is partitioned according to the number of case attributes involved: earlier phases include rules with fewer attributes, while later phases expand to incorporate more. For example, the initial sub-knowledge bases contain rules for basic case attributes, such as whether the defendant is a repeat offender. In contrast, later phases include rules incorporating additional factors, such as the defendant's cooperation with law enforcement, whether the crime involves mitigating circumstances, etc. Specifically, we define the sub-knowledge base for phase $p$ as:

$$
\mathcal{KB}_p = \left\{ y = (1 + w^\top z)(am + b) \mid \mathbf{1}^\top z \le p \right\}, \tag{17}
$$

ensuring that earlier phases focus on simpler cases, thereby facilitating a smoother and more stable training process.

