# OpenReview forum: "Curriculum Abductive Learning"
_NeurIPS.cc/2025/Conference — NeurIPS 2025 poster_

### Official Review · Reviewer_pZZ6 · 2025-06-17

**Clarity:** 3
**Significance:** 2
**Originality:** 3
**Rating:** 4
**Confidence:** 3

**Summary:**

This paper proposes C-ABL, a curriculum-based extension to abductive learning (ABL), aiming to mitigate the large reasoning space and unstable supervision problems faced by standard ABL frameworks. Specifically, C-ABL partitions the logic knowledge base into structured subsets according to dependency relations among logical rules and progressively introduces these subsets into training, facilitating more stable and efficient learning. Empirical results on arithmetic tasks, chess attack prediction, and judicial sentencing demonstrate improved accuracy and faster convergence compared to baseline ABL methods.

**Questions:**

I should first acknowledge that I am not an expert in ABL, but I believe I have understood most of the paper and have the following questions. If my understanding is incorrect, please feel free to clarify—I would be happy to raise my score.

---

### 1. Data partitioning relies on true concept labels—is this a form of information leakage?

As I understand it, once the knowledge base is partitioned into curriculum phases, the training data is also filtered accordingly—only examples whose **concept labels lie entirely within the current Zₚ** are used in phase p. But to do this, one must know the true concept labels of each sample in advance. So even though the model is never directly given these labels during training, the data filtering procedure still **relies on ground-truth concept labels**, which the baselines (ABL, A³BL) do not use. This creates an imbalance.

- Could you explain why this setup does not introduce information leakage or unfair advantage?
- Have you tried giving the same phase-specific training data streams to the baselines to ensure a fair comparison?

---

### 2. Is curriculum learning really necessary?

Even setting aside the above concern, I still have doubts about whether curriculum learning is actually needed. For example, in the digit addition task, what would happen if we train **only in phase 1**, not moving to later phases when performance exceeds random, but simply continuing phase 1 training until convergence, then evaluating? After all, we’re only training the perception model. If it has learned to predict digits well, then at test time we could just pair it with the **full KB** and run inference—do we really need to go through all phases?

- Have you tried this “phase-1-only” setting and compared its performance or convergence time?
- I can imagine that training only on phase 1 might limit the number of usable samples, but couldn’t we mitigate that by generating more phase 1 data? Given that the task is relatively simple, I’m not convinced that multi-phase curriculum is essential.

I'd appreciate clarification on these points—they are important to my evaluation of whether the gains come from the curriculum design itself, or from more subtle data handling choices.

**Ethical Concerns:**

["NO or VERY MINOR ethics concerns only"]

**Final Justification:**

The author's response has effectively alleviated my concerns about data leakage and the necessity of curriculum learning.

**Limitations:**

yes

**Quality:**

2

**Strengths And Weaknesses:**

Strengths

1. The proposed idea of introducing a logic-guided curriculum into abductive learning is interesting and well-motivated.

2. The paper is clearly written and easy to follow.

3. Experimental results demonstrate significant improvements in accuracy and training efficiency compared to prior methods.

Weaknesses

 1. I still have serious concerns and uncertainties about the soundness of the proposed algorithm, especially regarding the fairness and potential information leakage introduced by the curriculum design (see detailed questions below).

---

> ### Author Rebuttal · Authors · 2025-07-30
>
> Thank you for your supportive and insightful comment. We will address your concerns as follows:
>
> ---
>
> **Q1**  Why this setup does not introduce information leakage or unfair advantage? Can you give the same phase-specific training data streams to the baselines to ensure a fair comparison?
>
> **A1**  First, we would like to emphasize that the machine learning model $f$ in C-ABL does not access ground-truth concept labels during training. On one hand, as you noted, the supervision signal for $f$ still comes solely from the target label $y$; on the other hand, each sub-base in C-ABL often contains multiple logically related concepts rather than focusing on a single one, so the process will not reduce to supervised learning of individual concepts and will not diminish the effect of reasoning shortcuts. The partitioning is performed before training as part of the curriculum design, and the concept labels are never exposed to $f$ or used for its supervision—exactly as in the baselines—ensuring that no information leakage occurs.
>
> To ensure fairness, we also trained the baselines using the same phase-specific data streams on the MNIST addition (hexadecimal) task, and the final accuracies are reported below. As seen, simply applying the same schedule to the data without partitioning the knowledge base leads to very poor performance. This is primarily because unrelated or noisy rules can produce incorrect pseudo-labels, which severely undermines both learning stability and final accuracy.
>
> |                        | d = 1 | d = 2 | d = 3 |
> | ---------------------- | ----- | ----- | ----- |
> | ABL (data scheduling)  | 17.58 | 15.62 | 10.30 |
> | A³BL (data scheduling) | 19.45 | 14.95 | 12.98 |
> | C-ABL                  | 64.25 | 64.25 | 66.67 |
>
> The above findings demonstrate that **the main performance gain of C-ABL stems from the principled partitioning of the knowledge base**, which forms the foundation of our curriculum approach. Without this structured logic-level partitioning, no scheduling strategy alone would deliver the observed improvements.
>
> ---
>
> **Q2**  Is curriculum learning really necessary? What would happen if we train only in phase 1？
>
> **A2**  Thank you for the thoughtful question. Following your suggestion, we conducted an additional ablation experiment on the “phase-1-only” setting (with more data generated but only partial knowledge base $\mathcal{KB}_1$). As expected, the model trained solely on phase 1 converges more quickly in the early stages but exhibits extremely poor final performance. We will include this ablation study, along with full results and convergence curves, in the appendix of the final version. Briefly: on the MNIST Addition task (hexadecimal, $d=3$), test accuracy drops to **21.26%** (C-ABL: 66.67%); on Chess Attack, to **12.59%** (C-ABL: 86.79%); and on Legal Judgement Prediction, the F1 score falls to **0.325** (C-ABL: 0.904).
>
> The underlying reason is that **the knowledge base in phase 1 is structurally incomplete**—it omits essential rules required to handle the full task (e.g., necessary legal rules for handling more comprehensive sentencing scenarios). Simply generating more phase-1 data cannot compensate for this, as the model is never exposed to the full set of reasoning patterns. These findings further confirm that a multi-phase curriculum is necessary for progressively introducing the complete rule structure and enabling generalizable learning.
>
> ---
>
> Thank you again for recognizing the clarity, motivation, and empirical strength of our paper. We will reflect these revisions in the final version of our paper.

---

### Official Review · Reviewer_c9eA · 2025-06-27

**Clarity:** 4
**Significance:** 1
**Originality:** 3
**Rating:** 3
**Confidence:** 3

**Summary:**

This paper presents a framework for curriculum learning for abductive learning.

Abductive learning (ABL) is a domain where a neural network is trained in tandem with a knowledge base (KB) to learn to recognize concepts from inputs, where the supervision comes from determining which concepts can be combined with rules from the KB to produce the supervised labels. To train a model, existing ABL training frameworks are applied to each KB in the curriculum until above-chance accuracy is reached for all concepts.

The paper gives an algorithm for constructing a curriculum of increasingly large KB's up to the original KB based on a dependency graph of rules. It also gives a theoretical analysis of how a curriculum, by reducing the size of the the abduction space at each training phase, can speed up convergence of ABL.

The paper presents experiments on d-digit addition (given MNIST images, rules of addition, and sum outputs, learn to recognize digits and predict correct sums of them), chess attack (given images of chess boards, rules of piece attacks, learn to recognize which pieces are attacking other pieces), and judicial sentencing (given criminal judgement records, predict the sentencing factors. The new method learns faster and is more accurate.

**Questions:**

For chess attack, how do you know where the pieces are located? It looks like the only concepts that then neural network predicts is which piece is in the image. I don't understand this at all. Does the neural network take in one image, and predict multiple things, or is it applied separately to multiple images? Can you fully describe the input space, the concept space, the rule set, and the supervised labels for chess attack? It would probably be good to do this in the appendix for each of the tasks. In the problem setting, the model f is described as mapping each input element to a concept label, but I don't understand how this works for chess attack or judicial sentencing.

What does "pre-training setting" mean for judicial sentencing?

My main reservation with this paper is about whether this type of research is significant. A discussion on the doubts raised above that convinces me that this line of research is relevant — either showing how current ABL research can have applications or is elsehow useful to research, or conveying how ABL research gives insight to related fields, or another way of justifying the significance of the research) would convince me to increase my score.

**Ethical Concerns:**

["NO or VERY MINOR ethics concerns only"]

**Final Justification:**

It is clear this paper is borderline, but it is hard to know whether it should be borderline accept or borderline reject. I do not have high confidence that I have the best perspective of the reviewers, but I will keep my score as the author rebuttal did not change my mind much about the significance of this line of research. If this concern were addressed, I would raise my score to a 4 or 5.

**Quality:**

4

**Strengths And Weaknesses:**

## Summary of strengths and weaknesses
The paper is really well written and has strong results. The results are incremental and predictable (applying curriculum learning to a niche neurosymbolic domain that hasn't seen curriculum learning of this type before) but also very solid. My biggest reservation is that the area of research seems to be a toy neurosymbolic domain that is outdated (feels like a pre-2020 type of AI research paper) and is not very significant of a research direction for AI in 2025.

### Strengths
- The writing, paper organization, description of methods and experiments, and discussion of results are great. The writing is clear and easy to understand.
- The approach is simple and easy to understand. It is well-designed and described well with pseudocode and text. Helpful clarifying examples are included.
- The theoretical analysis is relevant and makes sense at a high level. It is a nice cherry on top to justify the curriculum approach.
- The paper does a good job briefly reviewing abductive learning for unfamiliar readers, and setting up preliminaries with knowledge base concepts, etc.
- The approach achieves higher accuracy and faster training time for the evaluated domains.
- Overall, the paper is polished, dots its i's and crosses its t's, and is good.
- It's nice that the curriculum can be automatically constructed from a knowledge base.
- (Originality) from what I can tell, the proposed approach has not been applied to ABL before.

## Weaknesses
- It is an incremental paper (Add on curriculum learning to existing ABL training schemes to improve performance). It is not very surprising that curriculum learning can improve training speed and final performance of the described system.
- Moreover, the novel proposals in the paper are specific to abductive learning. I don't think there is any insight for domains besides abductive learning, since curriculum learning, as a broad technique, has been applied to a ton of different domains.
- Abductive learning is a pretty niche problem field. It seems like a toy neurosymbolic problem that is academically studied by a few people, but doesn't have very much real world potential or relevance for AI.
- Abductive learning and the accompanying neurosymbolic training setups, with the assumptions about what types of data, supervision, and knowledge bases are available for a problem, seems extremely outdated and irrelevant for modern AI.
- All of the problems studied are contrived setups that only work for research works studying abductive learning. For example, two of the three domains use MNIST images for the perceptual component. All use a logical knowledge base encoding of the rules for each domain.  This type of data isn't that prevalent and especially this combination logical rule systems, simple perception, and then learning perception from the logical rules is academically interesting, but doesn't seem to have any real world usefulness.
- LLM's seem like a far better way to approach this problem — first learn perception, then learn to reason given existing perception. I think the premise of abductive learning is fundamentally limited.

##  Further reflection on the state of neurosymbolic research

The main weakness of this paper is that neurosymbolic abductive learning doesn't seem like a very significant, impactful, or important field of computer science research. Neurosymbolic AI has been a popular field of research for the past decade+, and there have been many subareas of research on how to combine neural and symbolic systems for learning and reasoning. Earlier on, there was more "things are wide open" for how future AI systems would solve the tasks we're concerned about. Especially in the late 2010's, there was a commonly held sense of the inability for purely neural systems to deal with generalization, compositionality, reasoning, logical rules, and the like. Abductive learning was one such direction, and primarily concerns itself with how reasoning systems can provide learning signal to a perceptual system. I was once heavily immersed in neurosymbolic research of similar varieties.

However, as machine learning and especially NLP and LLM research has developed and progressed, a lot of these neurosymbolic methods have become obsolete.

Neurosymbolic AI is not completely obsolete — it has evolved and there are always new ways to usefully combine neural and symbolic systems. There's really exciting work about generating code to reason (ViperGPT), AlphaGeometry which does theorem proving with a LLM and a symbolic rule system, and many other directions too. However, I think the problem setting that abductive learning is built on is fundamentally outdated and not going to provide useful AI research.

---

> ### Author Rebuttal · Authors · 2025-07-30
>
> Thank you for your supportive and insightful comment. We will address your concerns as follows:
>
> ---
>
> **Q1**   What is the significance of your paper?
>
> **A1**  While curriculum learning has indeed been widely applied, **our work is the first to design a curriculum from a logic-level perspective**, by explicitly partitioning the knowledge base according to logical dependencies and complexity. This is not a trivial extension of standard curriculum strategies that merely reorder data samples or tasks. Our partitioning algorithm (Algorithm 1) formalizes how to decompose a rule set into sub-bases that enable smoother abductive reasoning and provable reductions in the search space (Theorem 4.2), which is a novel insight applicable to any neuro-symbolic framework requiring structured reasoning. In this sense, our approach **provides a general methodology for curriculum design in knowledge-driven systems**, potentially extending beyond abductive learning.
>
> ---
>
> **Q2**   Why is ABL/NeSy research significant and relevant?
>
> **A2**  We appreciate the reviewer’s perspective on the evolution of neuro-symbolic AI. It is true that recent advances—particularly in LLMs and code-generating systems like ViperGPT and AlphaGeometry—have opened exciting new paths for integrating symbolic reasoning. However, we believe that abductive learning (ABL) retains distinct value, especially in domains where **explicit, verifiable, and logic-consistent reasoning remains critical**.
>
> ABL differs from many other paradigms in that it maintains the **full formal rigor of symbolic reasoning**, rather than relying on implicit heuristics or approximations. This makes it especially well-suited for real-world domains that possess both data and well-curated expert knowledge—such as judicial sentencing, historical document understanding, medical diagnosis, etc.—where knowledge bases are often **readily accessible and incorporates a wealth of valuable information**, yet **the challenge lies in leveraging this rigorous and insightful domain knowledge effectively**. ABL methods, explicitly driven by both data and knowledge without sacrificing integrity of each side, have demonstrated strong results in such domains [1,2,3,4]—an example is also given in our legal sentencing task (Sec. 5.3). LLM-based methods, while powerful, typically rely on implicit mechanisms (e.g., retrieval or prompting), which fall short in fully exploiting structured knowledge and lack transparency.
>
> Moreover, in safety-critical domains like industrial control or defense, ensuring strict adherence to rules is not optional. ABL and related logic-based frameworks enforce such constraints by design [5,6], making them suitable for **trustworthy, interpretable, and legally or ethically constrained AI**—an area where black-box models remain insufficient.
>
> Our contribution in this paper—C-ABL—advances this line of research by introducing the first curriculum framework for ABL that strategically organizes symbolic knowledge over learning phases. This allows reasoning complexity to grow in a controlled, data-efficient, and semantically stable way, further broadening the practical applicability of ABL systems, and potentially applicable to a wide range of neuro-symbolic paradigms.
>
> ---
>
> **Q3**  Further clarification on experiment of chess attack and judicial sentencing.
>
> **A3**  Thank you for the question and the opportunity to clarify. In the chess attack task, the positions of the pieces are given and fixed as part of the input layout, and the learning model $f$ is applied to each individual board cell image to identify what piece is present at that location. The concept label $z_i$ corresponds to the identity of the piece in cell $i$, not its position. Each input $x_i$ is a small image of one board cell, and $f$ maps it to a concept label $z_i$. These predicted concepts are assembled into a board-level representation, and logical rules (e.g., for attack range) are then applied to determine if the king is under threat. We will make this clearer in the final version of the paper.
>
> In the judicial sentencing task, as described in Lines 323–325 and Appendix F.3, the model $f$ classifies whether the criminal jugement record expresses certain sentencing factor (e.g., “voluntary surrender”), and then reasoning is performed over these predicted factors using the legal knowledge base.
>
> For the "pre-training setting", thank you for pointing this out. The term may have been used imprecisely in the original draft. Consistent with the protocol in the original ICDM 2020 paper [1], we assume that a small portion of the dataset (e.g., 10% or 50%) includes ground-truth concept labels. In addition to the abductive learning process, the model also learns to predict concepts from this limited supervision. We will revise the text to clarify this terminology and avoid potential confusion.
>
> ---
>
> Thank you again for your high recognition of the quality, clarity and originality of our paper.
>
> ---
>
> [1] Huang et al. "Semi-Supervised Abductive Learning and Its Application to Theft Judicial Sentencing." ICDM 2020.
>
> [2] Gao et al. "Knowledge-Enhanced Historical Document Segmentation and Recognition." AAAI 2024.
>
> [3] Wu et al. "Abductive Multi-instance Multi-label Learning for Periodontal Disease Classification with Prior Domain Knowledge." Medical Image Analysis 2025.
>
> [4] Hu et al. "Efficient Rectification of Neuro-Symbolic Reasoning Inconsistencies by Abductive Reflection." AAAI 2025 (AAAI outstanding paper).
>
> [5] Hoernle et al. "MultiplexNet: Towards Fully Satisfied Logical Constraints in Neural Networks." AAAI 2022.
>
> [6] Belle. "On the Relevance of Logic for AI, and the Promise of Neuro-Symbolic Learning" Neurosymbolic Artificial Intelligence 2025.

---

> > ### Comment · Reviewer_c9eA · 2025-08-01
> >
> > Thank you for your professional response and for addressing my concerns. I acknowledge that while I have experience with a range of neurosymbolic methods, I'm not an expert on abductive learning and the details of these domains. Clearly, the other two reviewers have more expertise in this narrow domain. At the same time, the problems still seem to be very toy problems (as your explanation of the chess attack problem further convinces me.
> >
> > I will defer to the other reviewers expertise for the most part, but am happy to participate in the discussion to provide commentary as needed.

---

> ### Author Response · Authors · 2025-08-03
>
> Thank you for your thoughtful follow-up. While we agree that some ABL tasks, including some used in our work, may appear simplified, we believe the core idea—designing a curriculum through knowledge base partitioning to reduce reasoning complexity—has broader implications and can benefit a wide range of knowledge-driven AI systems and tasks.
>
> We truly appreciate your high recognition of our paper’s quality, clarity, and originality. Please feel free to reach out if further discussion would be helpful.

---

### Official Review · Reviewer_dk43 · 2025-07-02

**Clarity:** 3
**Significance:** 2
**Originality:** 3
**Rating:** 5
**Confidence:** 4

**Summary:**

The authors study the neurosymbolic learning problem of learning a concept-based neural network with logical supervision. They extend 'abductive learning', which finds the most likely logically consistent prediction as a weak supervision label. This search process is hard to scale, and can introduce instabilities. Therefore, the authors design a curriculum schedule, making the problem progressively harder. They do this via a sequence of increasingly more granular knowledge bases. This results in a more scalable and better performing method.

**Questions:**

**Critical** (if these are properly answered I'd be happy to increase my score)
- The paper suggest they make use of concept labels during training, which (usually) is not done in the baselines.
    - Lines 191-195 suggest the authors make use of concept labels to determine when to proceed to the next phase. Most baselines assume no access to concept labels. If indeed the authors use these concept labels, they significantly collapse the number of reasoning shortcuts in the problem [1], making it much easier than the problem the baselines have to solve. If this is indeed what is done, I do not believe the experimental results are a valid comparison. If so: Please rerun the method without concept labels, or rerun the baselines with (this limited bit of) concept supervision.
- Theorem 4.2 has an unlisted assumption that possibly impacts the design of the method. This should be clarified.
   - The proof of theorem 4.2 relies on the fact that the consistency on concepts in the previous phase remains the same. However, during training (especially if the dataset changes), there is no guarantee (or proof) that this ordering remains valid. Then the search collapses to the exponential bound given in the lemma.
    - Line 234 suggests the search is only done over the new concepts. Is this correct? If so, is this indeed still valid, given the concern above?
- Theoretical aspects and details that should be clarified:
    - The domain knowledge is said to be in first-order logic. Looking at the examples in the appendix, it looks like the authors use a logic programming formalism, maybe equivalent to DataLog? The semantics should be more clearly specified, and it does not look (at least in terms of how it's used in practice) to be equivalent to a full FOL which allows infinite domains. Furthermore, the definition in 2.1 is insufficient for interpreting the rules.
    - Section 3.1: The algorithm does not assume acyclic graphs, right? How does it handle cycles?
    - Algorithm 1: It should be clarified how the topological sorting works. The precedence ordering is only a partial order (I think) so topological sorting is not unique. Also, the precedence ordering is a heuristic, right?
    - How is $\tau$ optional? Its role is not very clear.
    - Line 189: Can you make this scheduling more precise? What about data labels $y$ that do not appear in the current phase?

**Minor points & questions**:
- I really like the explanation of abductive learning, which is minimal and clean. It also made me realise the connection to the top-k inference method common in eg Scallop [2] and other approximate inference methods [3]. Where abductive learning uses $k=1$ (ie: use the MAP state as supervision). Are there any benefits for doing this instead of using weighted model counting, like in eg DeepProbLog? Maybe scalability, although there are also recent efforts on improving this that go beyond what C-ABL is tested on (see eg [4-10]). In particular, [4, 5, 9, 10] also look into decomposing the knowledge base in various methods.
- Line 114-115: "For each input x, the correct concept labels should be unique". This is an assumption, not something that one can state. See Assumption **A1** in [1]
- In table 1, please specify this is about concept accuracy, not sum prediction accuracy (right?)
- Line 336-337: These probabilistic logic programming languages are indeed tied to specific logic paradigms, but so is C-ABL, right? There are also probabilistic methods that do not assume a particular language, see eg [1, 4, 5, 6, 8, 9]


1. Marconato, Emanuele, et al. "Not all neuro-symbolic concepts are created equal: Analysis and mitigation of reasoning shortcuts." Advances in Neural Information Processing Systems 36 (2023): 72507-72539.
2. Li, Ziyang, Jiani Huang, and Mayur Naik. "Scallop: A language for neurosymbolic programming." Proceedings of the ACM on Programming Languages 7.PLDI (2023): 1463-1487.
3. Manhaeve, Robin, Giuseppe Marra, and Luc De Raedt. "Approximate inference for neural probabilistic logic programming." Proceedings of the 18th International Conference on Principles of Knowledge Representation and Reasoning. IJCAI Organization, 2021.
4. van Krieken, Emile, et al. "A-nesi: A scalable approximate method for probabilistic neurosymbolic inference." Advances in Neural Information Processing Systems 36 (2023): 24586-24609.
5. van Krieken, Emile, et al. "Neurosymbolic Diffusion Models." arXiv preprint arXiv:2505.13138 (2025).
6. De Smet, Lennert, Emanuele Sansone, and Pedro Zuidberg Dos Martires. "Differentiable sampling of categorical distributions using the catlog-derivative trick." Advances in Neural Information Processing Systems 36 (2023): 30416-30428.
7. De Smet, Lennert, and Pedro Zuidberg Dos Martires. "A Fast Convoluted Story: Scaling Probabilistic Inference for Integer Arithmetic." arXiv preprint arXiv:2410.12389 (2024).
8. Verreet, Victor, et al. "EXPLAIN, AGREE, LEARN: Scaling Learning for Neural Probabilistic Logic." arXiv preprint arXiv:2408.08133 (2024).
9. Choi, Seewon, et al. "CTSketch: Compositional Tensor Sketching for Scalable Neurosymbolic Learning." arXiv preprint arXiv:2503.24123 (2025).
10. Ahmed, Kareem, Kai-Wei Chang, and Guy Van den Broeck. "Semantic strengthening of neuro-symbolic learning." International Conference on Artificial Intelligence and Statistics. PMLR, 2023.

**Ethical Concerns:**

["NO or VERY MINOR ethics concerns only"]

**Final Justification:**

This is quite a solid paper with a nice contribution that can be used in several areas of Neurosymbolic AI. I had a clear discussion with the authors who seemed to have an open mind, and they resolved my main concerns (in particular, the use of validation concepts in phase transitions).

**Limitations:**

Limitations are listed in the appendix.

**Quality:**

3

**Strengths And Weaknesses:**

Strengths:
- The paper is clearly written
- The idea introduced is interesting and novel as far as I know. It is also theoretically well motivated, and may be extended to other NeSy methods.
- Experimental results are quite strong

Weaknesses:
- I have a few concerns about some of the theory and validity of comparisons in terms of experimental setup.
- Scalability of the method remains unproven compared to the state of the art in the field.

---

> ### Author Rebuttal · Authors · 2025-07-30
>
> Thank you for your supportive and insightful comment. We will address your concerns as follows:
>
> ---
>
> **Q1** Does the method use concept labels during training?
>
> **A1**  To enable consistent transitioning across experiments, the phase transition check described in Lines 191–192 indeed involves access to concept label information. Yet it is worth noting that during experiments, this check is performed exclusively on a validation set, and the concept labels are not used during training or backpropagation.
>
> Thank you for pointing this out, and, as a further exploration, we **re-ran the MNIST Addition** (hexadecimal) experiment using a fixed-interval schedule that **do not require access to concept labels**—specifically, the transition to the next phase occurs every 200 iterations. As shown below (each cell reports test accuracy and training time), C-ABL still maintains a clear advantage over the baselines, demonstrating that the improvements arise from the structured curriculum itself rather than from any concept label usage.
>
> |                                   | d = 1        | d = 2        | d = 3         |
> | --------------------------------- | ------------ | ------------ | ------------- |
> | ABL                               | 60.87 (20.8) | 61.75 (49.1) | 64.14 (99.7)  |
> | A³BL                              | 22.45 (31.2) | 60.75 (68.2) | 65.81 (109.8) |
> | C-ABL                             | 64.25 (16.4) | 64.25 (32.0) | 66.67 (55.5)  |
> | C-ABL (fixed-interval transition) | 64.68 (21.9) | 64.59 (45.5) | 66.70 (70.9)  |
>
> Finally, we would like to emphasize that the learning model $f$ in C-ABL never accesses ground-truth concept labels during training: the only supervision signal comes from the task label $y$, just as in the baselines. The validation-based threshold is used solely for determining when to trigger phase transitions, and does not introduce information leakage into the learning process.
>
> ---
>
> **Q2**  What assumptions underlie Theorem 4.2? How is the stability of previous phase predictions ensured in Theorem 4.2?
>
> **A2**  We agree with you that Theorem 4.2 relies on the stability of concepts learned in previous phases. In our method, this stability is ensured by the curriculum schedule (Lines 191–195), where a new phase only begins once $z \in \mathcal{Z}_p$ achieve accuracy above the random baseline, providing a sufficiently reliable foundation. Empirical evidence in Figure 3 (and Appendix F.1) shows that earlier concepts retain their accuracy during later training, validating this assumption in practice.
>
> Moreover, as is common in neuro-symbolic learning, we assume that the semantics of concepts remain constant across the dataset [1,2], characterized as $p(z\mid x)$ remain unchanged as the dataset or knowledge base evolves. This is a natural and implicit assumption in many settings—for example, an image of the digit three consistently corresponds to the concept $z = \texttt{three}$ regardless of training phase. It enables C-ABL to build on previously learned reasoning without semantic drift. We appreciate your comment on this point and will revise Theorem 4.2 to explicitly state this assumption and clarify why the abduction space reduction remains valid and does not revert to the exponential bound in Lemma 4.1.
>
> ---
>
> **Q3**  What is the logical formalism used to represent domain knowledge?
>
> **A3**  The domain knowledge, as in ABL or DeepProbLog, is indeed expressed in a logic programming formalism, implemented using Prolog, which is a well-known subset of first-order logic focusing on Horn clauses. While we use the term “first-order logic” broadly to emphasize its logical expressiveness, we agree that specifying the logic programming would improve clarity, and we will revise Section 2.1 accordingly in the final version of the paper.
>
> ---
>
> **Q4**   Details about the knowledge base partitioning algorithm.
>
> **A4**  We address below several questions regarding the partitioning algorithm (Algorithm 1). Thank you for raising these points—clarifications will be added in the final version.
>
> - (Cycles) You are correct that our algorithm does not assume acyclic dependency graphs. In cases where the concept dependency graph contains cycles, Line 4 of Algorithm 1 typically groups the involved rules into the same sub-base, which is then introduced as a whole in the curriculum. However, in practice, the knowledge bases rarely exhibit cycles during the initial graph construction—a cycle would imply that the conclusion of one rule is also used to justify its own premise, which is uncommon.
>
> - (Topological sort) Regarding the topological sort in Algorithm 1 Line 8, we acknowledge that the original explanation may have been too brief due to space limitations; below is a more detailed clarification. The ordering is derived as follows: if there exists an edge $(r_i, r_j) \in E$ with $r_i \in C_a$ and $r_j \in C_b$, then we set $C_a \prec C_b$. This reflects that sub-base $C_b$ depends on $C_a$, and should appear later in the curriculum. Since this forms only a partial order, multiple valid topological sorts are possible; for unordered pairs, their relative order has no effect on the algorithm’s behavior—just like in standard topological sorting procedures, where such pairs are typically ordered arbitrarily. In our implementation, we heuristically place smaller clusters earlier, but we have observed almost identical experimental results when using a random ordering. We will add this in the final version of the paper.
>
> - (Hyperparameter $\tau$) As for the use of $\tau$, as stated in Line 167, it serves to avoid overly fine-grained partitions by allowing small clusters to be optionally merged with their immediate successors until a minimum number of concept predicates per phase (namely, $\tau$) is reached. This parameter is optional—if one does not mind having a larger number of phases, it may be omitted. We provide a sensitivity analysis of $\tau$ in Table 4 of Appendix F for reference.
>
> ---
>
> **Q5**  Connection between ABL and top-k inference.
>
> **A5**  Regarding the choice of using the MAP state (i.e., $k=1$) as supervision, this design is indeed intentional. First, it provides computational advantages over approaches like weighted model counting, as it avoids the combinatorial cost of summing over all possible explanations—particularly valuable in early training when many candidates are noisy or semantically implausible. Second, using a single best explanation greatly simplifies the abduction process, making it more tractable to identify, validate, and refine hypotheses throughout training.
>
> We appreciate your references to recent work on probabilistic inference methods, and will update the discussion in the final version to better contextualize our approach within this broader landscape.
>
> ---
>
> **Q6**  Other minor points.
>
> **A6**  Thank you for the thoughtful comments and references—we appreciate the reviewer’s broad perspective and insightful connections across related work.
>
> - (Unique assumption) We will revise Line 114–115 to clarify that the uniqueness of concept labels per input is an assumed property of the datasets, and we will reference Marconato, Emanuele, et al., 2023 to contextualize this more carefully.
>
> - (Accuracy) Table 1 indeed reports concept prediction accuracy, as in ABL and A³BL—we will make this explicit in the caption to avoid ambiguity.
>
> - (Related work) Regarding Lines 336–337, we will clarify that C-ABL primarily uses first-order logic, which, while specific, remains expressive enough to capture a wide range of domain knowledge. Additionally, according to the official code of ABL [3], ABL framework can also be implemented with other logical formalisms, e.g., propositional logic and satisfiability modulo theories; similarly, C-ABL can be adapted to these settings with minor modifications.
>
> ---
>
> Thank you again for your recognition of our paper. We also sincerely appreciate your review comments, which have provided us with valuable new insights. We will incorporate the corresponding revisions in the final version.
>
> ---
>
> [1] Marconato et al. "Neuro-Symbolic Continual Learning: Knowledge, Reasoning Shortcuts and Concept Rehearsal." ICML 2023.
>
> [2] Marconato et al. "BEARS: Make Neuro-Symbolic Models Aware of their Reasoning Shortcuts." UAI 2024.
>
> [3] Huang et al. "ABLkit: A Python Toolkit for Abductive Learning." Frontiers of Computer Science, 2024.

---

> > ### Comment · Reviewer_dk43 · 2025-08-01
> >
> > I thank the authors for their extensive response, which increased my confidence in the quality of the paper. A few follow-up thoughts:
> >
> > **A1**
> >
> > Thanks for running this. It is encouraging to see that the performance barely drops under this change. I think the claim that this method "does not introduce information leakage into the learning process", however, is unfounded, especially in tasks containing reasoning shortcuts. Also, I believe the manuscript should be clearer about the fact the method uses validation concept labels rather than training concept labels. I don't think this is _entirely_ correct practice, as I assume the hyperparameter tuning is also done on validation. However, it is quite minor.
> >
> > **A2**
> >
> > Thanks, this makes sense.
> >
> > **A3**
> >
> > This is somewhat imprecise, Prolog is not a subset of FOL, but also the presented paper does not implement full Prolog (unless I'm missing something). It doesn't implement negation in bodies, procedural predicates and function symbols, hence why DataLog seems more applicable.
> >
> > **A4**
> >
> > 1. I see, that makes sense. Note that cycles can be quite common for problems involving recursion (eg, the definition of a path).
> >
> > **A5**
> >
> > Thank you. Although I'm a bit surprised about "particularly valuable in early training when many candidates are noisy or semantically implausible", as [1] (Figure 3b) found top-k inference is much harder at the beginning of training.
> >
> >
> > **Unanswered question**
> >
> > Line 189: Can you make this scheduling more precise? What about data labels $y$ that do not appear in the current phase?
> >
> > 1. Manhaeve, Robin, Giuseppe Marra, and Luc De Raedt. "Approximate inference for neural probabilistic logic programming." Proceedings of the 18th International Conference on Principles of Knowledge Representation and Reasoning. IJCAI Organization, 2021.

---

> > > ### Author Response · Authors · 2025-08-01
> > >
> > > Thank you for your thoughtful comments and valuable follow-up questions, which helped us clarify key aspects of our method and identify areas to further improve the presentation.
> > >
> > > **A1**  Thank you for pointing this out. We agree that the manuscript should more clearly state that validation concept labels are used for determining the phase transition threshold. As demonstrated, even replacing the threshold with a fixed-interval schedule without any concept labels yields similar performance trends, suggesting that the use of validation labels has negligible impact on the experimental results. We will clarify this point explicitly in the final version.
> > >
> > > **A3**  You are correct that the paper may have presented a simplified view. While our implementation is based on Prolog, and certain rules used in experiments may include expressive constructs such as negation, the logic employed generally aligns more closely with Datalog in structure and usage (consistent with the knowledge bases in previous frameworks such as ABL and DeepProbLog). We will revise the wording to better reflect this.
> > >
> > > **A4** Thanks for the helpful remark. Indeed, cycles are common in recursive definitions (e.g., path finding), and we agree this is an important consideration. Our current experiments generally do not contain such rules, but we will clarify in the final version that our method supports grouping cyclically dependent rules into the same phase, ensuring correctness in the presence of cycles.
> > >
> > > **Unanswered question**  Apologies — we previously misunderstood this question as referring to phase transitions. As is common in curriculum learning, training examples that depend on concepts not yet supported by the current phase are temporarily deferred. Once the corresponding sub-base is introduced, these examples are included in training. In this way, if a target label $y$ cannot yet be supported by a valid abductive explanation under the current knowledge base, the corresponding sample will be skipped in that phase. We will clarify this behavior more precisely in the final version.
> > >
> > > We sincerely appreciate your engagement with our work and your insightful suggestions. We will incorporate the necessary clarifications into the final version to enhance the overall clarity and rigor of the paper.

---

> > > > ### Comment · Reviewer_dk43 · 2025-08-04
> > > >
> > > > I thank the authors and would like to confirm it clarified all my concerns. I'll raise my score under the assumption all clarifications are adapted in the manuscript.

---

> > > > > ### Author Response · Authors · 2025-08-04
> > > > >
> > > > > Thank you very much for your thoughtful and professional feedback. Your insights and well-chosen references have greatly helped us improve the clarity and rigor of our work. We appreciate your recognition and will ensure that clarifications are carefully incorporated into the final version of the manuscript.

---

### Official Review · Reviewer_7jdw · 2025-07-06

**Clarity:** 4
**Significance:** 3
**Originality:** 3
**Rating:** 5
**Confidence:** 4

**Summary:**

This paper proposes a novel method called Curriculum Abductive Learning (C-ABL), designed to address the instability and inefficiency of traditional Abductive Learning (ABL) when training under complex knowledge bases. C-ABL structurally partitions the knowledge base into sub-knowledge bases that are introduced progressively, enabling stage-wise model training. This approach reduces the hypothesis space and improves training efficiency. Theoretical analysis demonstrates that the method effectively lowers the reasoning complexity within each stage while ensuring smooth transitions between stages. Experiments across various tasks validate the superiority of C-ABL, showing significant improvements in accuracy, stability, and convergence speed.

**Questions:**

- Many real-world knowledge bases may lack clear modularity. How can the robustness of this method be validated in unstructured or noisy knowledge bases?
- Has the impact of different thresholds on final performance been experimentally quantified? Could certain tasks require more conservative/aggressive transition strategies?
- If new rules are added during training, does the knowledge base need to be repartitioned? Are there incremental partitioning solutions?
- Many domains use probabilistic logic. Can C-ABL accommodate uncertain reasoning? If not, which core assumptions need to be relaxed?

**Ethical Concerns:**

["NO or VERY MINOR ethics concerns only"]

**Final Justification:**

I recommend acceptance of this paper. The authors have not only addressed all major concerns but also provided substantial additional evidence through new experiments and theoretical elaborations that significantly strengthen the paper's contributions.

However, there are still some very small points:

1. While the probabilistic extension remains future work, the current deterministic framework already shows significant practical value in critical domains like judicial sentencing and medical diagnosis where rule-based certainty is paramount.

2. Some implementation aspects of Algorithm 1, particularly regarding cycle handling in dependency graphs and the τ threshold selection strategy (currently tested with values 2-5 in Appendix F), could benefit from more detailed exposition in the final version to facilitate reproducibility.

**Limitations:**

yes

**Paper Formatting Concerns:**

No major formatting issues.

**Quality:**

3

**Strengths And Weaknesses:**

#### **Strengths**
- It's the first work to explicitly leverage the internal logical structure of the knowledge base (KB) in ABL, rather than treating it as a black box, introducing phased sub-knowledge bases to dynamically manage reasoning complexity.

- Theorems 4.2 & 4.5 prove that C-ABL reduces hypothesis space size, drastically lowering iteration complexity. The method uses stone space and topological continuity analysis (Theorem 4.7) to ensure that no catastrophic forgetting during phase transitions. Sub-base partitioning adheres to logical completeness (Theorem 3.2), ensuring local reasoning conclusions remain valid under the full KB.

- Benchmarks against not only ABL variants (e.g., A$^{3}$BL) but also leading neuro-symbolic methods (NeurASP, DeepProbLog), highlighting the superiority of C-ABL in terms of training efficiency and stability.

- This work addressed ABL Core Issues. The curriculum design restricts candidate labels per phase, reducing erroneous supervision and the phased learning avoids model divergence among conflicting solutions.


#### **Weaknesses**:
- C-ABL requires the knowledge base to have modular or hierarchical structures. Otherwise, the partitioning algorithm (Algorithm 1) may fail, for example the highly interconnected rules and the random or flat rule sets.

- Current phase transitions rely on predefined accuracy thresholds,which may be unable to dynamically adjust phases based on real-time model performance and threshold settings may require task-specific tuning.

- While KB partitioning is offline, dependency graph construction complexity may become prohibitive for extremely large KBs.

---

> ### Author Rebuttal · Authors · 2025-07-30
>
> Thank you for your supportive and insightful comment. We will address your concerns:
>
> ---
>
> **Q1**  How can the robustness be validated in unstructured or noisy knowledge bases?
>
> **A1**  While C-ABL benefits from some degree of modularity, it does not assume perfectly clean or pre-modularized knowledge bases: It does not rely on manually specified modules or expert-defined clusters, instead, constructs a dependency graph based on actual logical use relations between predicates, and induces a curriculum in a structure-agnostic manner. It also allows fine-grained control via the threshold $\tau$ to avoid over-fragmentation in poorly structured KBs, therefore even when modularity is weak or noisy, C-ABL still yields meaningful phases by capturing local dependency clusters and avoids overfitting to artificial structure.
>
> In real-world tasks such as legal judgment prediction and chess attack, the knowledge bases we use are indeed not explicitly modular—many rules are loosely connected, overlapping, or even noisy—yet C-ABL still shows consistent improvements over ABL and A³BL, demonstrating its robustness in practice.
>
> ---
>
>
> **Q2**  Has the impact of different thresholds on final performance been experimentally quantified?
>
> **A2**  Our current transition strategy—advancing to the next phase once performance exceeds random—is already among the most aggressive. Even under this strict setting, C-ABL achieves strong results. We have also experimented with higher thresholds and observed that final test accuracy can improve slightly further. Thank you for the suggestion—designing more adaptive or fine-grained transition strategies is a promising direction we plan to explore.
>
> ---
>
> **Q3**  If new rules are added during training, does the knowledge base need to be repartitioned?
>
> **A3**  Thanks to the design of Algorithm 1 and the theoretical guarantee provided by Theorem 4.6, repartitioning is not necessary when new rules are added during training.
>
> ---
>
> **Q4**  Can C-ABL accommodate uncertain reasoning?
>
> **A4**  While the current version of C-ABL is built on deterministic abductive logic, its design does not preclude extension to probabilistic reasoning. Specifically, the core assumption that would need to be relaxed is the binary truth valuation used in standard logic programming (i.e., each concept is either true or false). To support probabilistic logic, one could extend the abductive engine to reason over distributions—e.g., by using weighted abduction or probabilistic logic programming frameworks, e.g.,  ProbLog or (perhaps) distributional extensions of ABL.
>
> The curriculum structure itself—i.e., the phase-wise introduction of logically related sub-bases—can remain intact. In fact, introducing probabilistic knowledge in a staged manner may help mitigate the complexity of inference. We consider such probabilistic C-ABL variants to be an exciting direction for future work.
>
> ---
>
> Thank you again for recognizing the overall ideas and significance of our paper.

---

> > ### Comment · Reviewer_7jdw · 2025-08-05
> > **Thank you for your response**
> >
> > I thank the authors for their response. Most of my questions have already been resolved. However, there are still some very small points:
> >
> > 1. While the probabilistic extension remains future work, the current deterministic framework already shows significant practical value in critical domains like judicial sentencing and medical diagnosis where rule-based certainty is paramount.
> >
> > 2. Some implementation aspects of Algorithm 1, particularly regarding cycle handling in dependency graphs and the τ threshold selection strategy (currently tested with values 2-5 in Appendix F), could benefit from more detailed exposition in the final version to facilitate reproducibility.

---

> > > ### Author Response · Authors · 2025-08-07
> > >
> > > Thank you for your follow-up and for recognizing the value of our framework in rule-critical domains. We agree that a clearer exposition of Algorithm 1 would enhance reproducibility. In the final version, we will elaborate on how cycles in the dependency graph are handled (e.g., typically by grouping mutually dependent rules), and clarify the role and impact of the $\tau$ threshold (as supported by the sensitivity analysis in Appendix F). We appreciate your thoughtful suggestions and will ensure these improvements are reflected.

---

### Decision · Program_Chairs · 2025-09-17

**Decision:**

Accept (poster)

**Comment:**

The paper initially received mixed reviews in the initial review with 1 accept, 1 borderline accept and 2 borderline reject scores. Major concerns raised were centered around three key aspects: the fairness and potential information leakage in the curriculum design, uncertainties and gaps in the theoretical assumptions and algorithmic details, and the limited empirical evaluation and broader relevance of the proposed approach.

The authors have provided a rebuttal that addresses many of these concerns, as evident from the increased scores to 2 Accepts, 1 Borderline Accept, and 1 Borderline Reject. The remaining concern, highlighted by reviewer c9eA, centers on the broader significance of abductive learning and its applicability to real-world tasks. The AC acknowledges this point, as it speaks to a valid and ongoing discussion within the neurosymbolic AI community regarding the evaluation of new methods on "toy" problems versus large-scale, real-world applications. However, the AC believes that this line of research does have potential for providing a way towards interpretable, logic-consistent systems that can benefit from symbolic systems. For these reasons, I recommend acceptance. The authors should incorporate the clarifications from their rebuttal into the final version of the paper.